



# Optical complexity of North Sea, Baltic Sea, and adjacent coastal and inland waters derived from satellite data

Martin Hieronymi[1], Daniel Behr[1], Shun Bi[1,2], Rüdiger Röttgers[1]

[1]Department Department of Optical Oceanography, Institute of Carbon Cycles, Helmholtz-Zentrum Hereon, 21502 Geesthacht, Germany
[2]State Key Laboratory of Lake Science and Environment, Nanjing Institute of Geography and Limnology, Chinese Academy of Sciences, Nanjing 210008, China

*Correspondence to*: Martin Hieronymi (martin.hieronymi@hereon.de)

**Abstract.** Despite advances in remote sensing, consistent monitoring of water quality across freshwater-marine systems remains challenging due to methodological fragmentation. Here, we provide an overview of an exemplary dataset on water quality characteristics in inland waters, coasts, and the open sea estimated from optical satellite data. Specifically, this is Sentinel-3 OLCI data for the entire North Sea and Baltic Sea region for the period June to September 2023. The dataset includes daily aggregated observational data with a spatial resolution of approximately 300 m of reflectance at the top-of-atmosphere

and for cloud-free water areas remote-sensing reflectance, inherent optical properties of the water, and an estimation of the concentrations of water constituents, e.g. related to the aquatic carbon content. These are the results of the novel A4O atmospheric correction and the ONNS water algorithm. The dataset serves as a prototype for understanding the processing chain and interdependencies, but also for developing a high degree of connectivity for answering various scientific questions; we do not perform an actual validation of the 73 individual parameters in the dataset. The aim is to show how fragmentation

in water quality monitoring along the aquatic continuum from lakes, rivers to the sea can be overcome by applying an optical water type-specific and neural network-based processing scheme for Copernicus satellite data. Emphasis of this work is on analysing the optical complexity of remote-sensing reflectance in the North Sea, Baltic Sea, coastal, and inland waters. Results of a new optical water type classification show that almost all (99.7%) remote-sensing reflectances delivered by A4O are classifiable and that the region exhibits the full range of optical water diversity. The dataset can serve as a blueprint for a

holistic view of the aquatic environment and is a step towards an observation-based digital twin component of the complex system.

## 1. Introduction

Satellite remote sensing enables a continuous and global observation of the Earth system and the interactions of its compartments. Optical remote sensing, which measures the radiance of reflected sunlight in the visible to near-infrared

spectrum, allows large-scale observation of water surfaces, enabling effective assessment of aquatic ecosystems across the freshwater-sea continuum. With an image swath width of 1270 km and a spatial resolution nadir of approximately 300 m, the Ocean and Land Colour Instruments (OLCI) on board two Sentinel-3 satellites are particularly well suited for global water



observation. The Sentinel-3 constellation operates on a near-polar and sun-synchronous orbit with images taken close to local noon. At full resolution, not only large marine areas but also complex coastlines, broad rivers, and larger inland waters can be

recorded. This enables the observation and evaluation of various issues, for example on drinking and bathing water quality, matter fluxes into the sea, the effects of human activities on the aquatic environment, and warnings of possible harmful algae blooms. However, different terminologies and inconsistent measurement practices are often used in limnology and oceanography. For a holistic view of the aquatic environment, it is therefore necessary to provide comprehensible and easily transferable reference values from satellite data. Moreover, for user-friendly usability, the data should not only be Findable,

Accessible, Interoperable, and Re-usable (FAIR), they should also be scientifically documented, contain a thorough metadata description, and enable easy data handling, e.g. through clear labelling and definition of limits and uncertainties. Ideally, the storage-intensive satellite images are also available in a simple visualisation so that their useability can be checked easily, as large land-sea surfaces are usually not visible due to clouds.

The view on different waters also poses significant challenges for analysing satellite images. This is mainly due to the optical

effect of the water constituents, which are parameterized by their inherent optical properties (IOPs). In addition to the pure water itself, phytoplankton, coloured dissolved organic matter (CDOM), and suspended non-algae particles (NAP) contribute to the colour appearance of the water body (e.g. Bi et al., 2023). Phytoplankton biomass is characterised by the total concentration of chlorophyll-a pigment ($Chl$ in mg m$^{-3}$). But the various algae species have different pigment compositions and occur in a wide range of colours (e.g. Xi et al., 2015; Lomas et al., 2024). CDOM is often a degradation product of

phytoplankton and organic compounds and is discharged from rivers into the sea in high concentrations but can also be brought into the sea by precipitation (e.g. Juhls et al., 2019; Kieber et al., 2006). The optical effect of CDOM is an absorption and darkening of the water. Sediments are suspended in shallow waters by currents, tides, and waves. Suspended particles increase the backscattering of the light in water and therefore result in relatively bright satellite image pixels. In addition to the water components mentioned, air bubbles, pollutants, and inelastic scattering processes can have an influence on the reflectance, but

this is neglected here. From the perspective of the satellite, however, the total signal from the water body is small compared to the atmosphere. And here too, the atmospheric properties and cloud conditions over inland waters and the ocean can vary considerably, e.g. due to land-originated dust or soot aerosols or maritime aerosols (e.g. Hess et al., 1998). The correction of all atmospheric influences, masking of clouds and provision of remote-sensing reflectance ($R_{rs}$ in sr$^{-1}$) valid for all water types is therefore a crucial step.

There are several methods for atmospheric correction (AC) and many water algorithms that derive water constituents from $R_{rs}$ (e.g. Müller et al., 2015; Brewin et al., 2015). The methods are usually optimized for different water bodies. However, no method offers robust performance across all water types and special atmospheric conditions, and the uncertainties and application limits are often insufficiently characterized; Hieronymi et al. (2023a), González Vilas et al. (2024), and others have documented this for various atmospheric correction methods for OLCI. The standard OLCI Level-2 (water full resolution)

baseline atmospheric correction is applied for all water surfaces but often delivers insufficient data quality for the Baltic Sea, many coastal, and inland waters, which is indicated, for example, by "negative" reflectance. In the Copernicus Marine Service,





different algorithms are used for the North Sea (as part of the North-East Atlantic) and the Baltic Sea (Brando et al., 2024), although this also includes inconsistent variables and designations; inland waters are not included here. Corresponding information for some lakes is provided in the Copernicus Land Service, but also with their own designations and completely

different data accessibility. For coastal marine waters, there are additional products with higher spatial resolution from Sentinel-2 MSI observations, but also with pixel-based algorithm switching (Brando et al., 2024). Thus, it should be the aim to cover all water areas uniformly and with adequate accuracy of all products.

Fuzzy logic classification of optical water types (OWT) quantitatively characterizes aquatic systems through continuous membership functions, overcoming the limitations of discrete classification schemes. This approach is particularly valuable

for capturing transitional waters (e.g., river plume or algal bloom fronts) where optical properties vary non-linearly with constituent concentrations. However, its effectiveness depends on valid $R_{rs}$ across the entire spectrum – a requirement often compromised when AC methods are optimized for specific water types. Hieronymi et al. (2023a) analyzed $R_{rs}$ from five AC with different OWT classification methods and found that the OWT algorithms have focused too much either on marine or limnological applications. This is expressed in the sometimes very low classifiability of $R_{rs}$ and the class assignment of spectral

shapes. The study also demonstrated the significant advantages of an integrated adaptation of atmospheric correction and OWT classification method, which also applies to the down-stream water algorithms. The findings from this study led to the development of a novel, robust, and comprehensive OWT classification system (Bi and Hieronymi, 2024).

In this paper, we document a dataset of merged Sentinel-3 OLCI Earth observations from the A4O-ONNS processing chain (Hieronymi et al., 2017), which includes the results of the new OWT classification. The individual processing steps have not

yet been adapted to each other and fundamental further developments will be implemented in the future. However, this dataset serves to visualise the overall system and to further develop the underlying algorithms, flags, and biogeo-optical relationships. This enables a systemic harmonization and optimization of the workflow, a validation of all water products and a more targeted estimation of the overall system uncertainties. The presentation of data from the current version of the algorithm enables feedback from users and aims to increase the data interoperability and FAIR-ness level; nonetheless, publication of the code

itself is only planned in the medium term. The aim is to show how fragmentation in water quality monitoring along the aquatic continuum from lakes, rivers to the sea can be overcome by applying an optical water type-specific and neural network-based processing scheme for Copernicus satellite data. Furthermore, in this work, some recommendations are realised to better characterize phytoplankton diversity and aquatic carbon fluxes in the future (e.g. Bracher et al., 2017; Brewin et al., 2023). The focus of this work is on the optical characterization of the waters of the North Sea – Baltic Sea region.




## 2. Data and methods

### 2.1. Geographical region

The spatial coverage of the Earth observation data extends from longitude -4.48° to 30.3° and latitude 48.98° to 65.9°, both in 0.004° steps (Fig. 1). This covers the entire Baltic Sea and North Sea with parts of the Norwegian Sea and English Channel, even parts of the Irish Sea are included with the city of Glasgow as a landmark for the western extension. The southern extent is defined by the inclusion of the entire Elbe River catchment. The spatial resolution of OLCI of approx. 300 m per pixel was tried to achieve for the overarching grid, resulting in an image size of 8695 x 4231 pixels in the coordinate reference system WGS 84 (EPSG:4326). Accordingly, smaller bodies of inland water and narrow rivers are not covered.

Four regional subdomains are defined for an initial categorisation of the optical diversity of the water bodies: North Sea, Baltic Sea, inland waters, and coastal waters; this is inspired by the domains in the Copernicus services (Brando et al., 2024). In terms of optical properties, the geographical subdomain "North Sea" is representative for the Copernicus Marine Service product for the Atlantic-European North-West Shelf which includes part of the Skagerrak in the east. In the Copernicus Marine Service, the Baltic Sea region is defined as an independent product, which also includes parts of the Skagerrak in the west at approx. 9° E. Correspondingly, a line at 9° E between Denmark and Norway is used to geographically separate the North Sea from the Baltic Sea. The inland waters subdomain contains all water areas on land that are larger than the pixel resolution of approx. 300 m, i.e. mainly lakes but also some broad rivers such as the Elbe estuary. The Copernicus Land Service offers Sentinel-3 OLCI-based[1] lake water quality products with a comparable resolution and additionally 100 m data from Sentinel-2 MSI[2]. In the optical characterisation of waters, the term "coastal waters" is often used; to define a subdomain for this, the 12 nautical miles zone of territorial coastal areas is used. In order to include brackish water areas of lagoons (e.g. the Curonian Lagoon), coastal lakes (e.g. the Ijsselmeer), and rivers influenced by the tide (e.g. the Lower Elbe), the coastal domain is defined as plus-minus 75 pixels along the coastal baseline (i.e. approx. ±22.5 km). The seaside zone roughly corresponds to the area of the high-resolution ocean colour product from the Copernicus Marine Service (with 100 m resolution from Sentinel-2 MSI[3]). The coastal waters therefore include parts of the other three domains. The images cover 36.8 million pixels in total, thereof are 61% the fraction of land, 27% North Sea, 11% Baltic Sea, and 1% inland waters. Coastal (or territorial) waters cover 13%.

---

[1] https://doi.org/10.2909/801137b8-9575-43ef-a073-140b663cc61c
[2] https://doi.org/10.2909/deae3db1-a214-4375-8d1a-63c42498050e
[3] North West Shelf Region: https://doi.org/10.48670/moi-00118, Baltic Sea: https://doi.org/10.48670/moi-00079





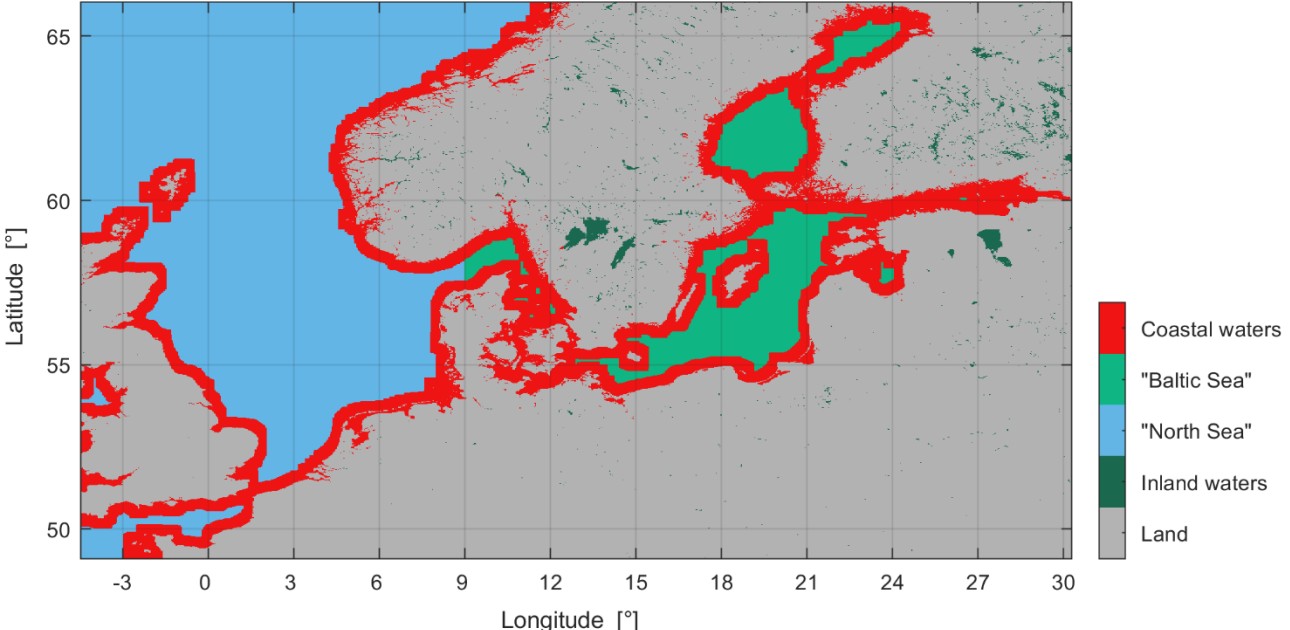


**Figure 1: Region of interest with definition of subdomains for optical water type analysis.**

## 2.2. Original Copernicus data

The freely available Sentinel-3A and 3B OLCI Level-1 data was obtained from the Copernicus Data Space Ecosystem (https://dataspace.copernicus.eu). All scenes with a contribution to the region of interest and within the period 1 June to 30

September 2023 were downloaded and processed. Among other things, the period covers a dedicated validation campaign in the Baltic Sea, where all water parameters from the satellite dataset were determined in-situ (Hieronymi et al., 2023b). Due to the polar orbit of the satellites, there are more frequent observation opportunities in the north and increasing gaps in the southern section (Fig. 2). The contributing images are taken around local noon, for this region within a time window of about three hours. It can therefore be said that the one-day aggregated data is representative for local noon.



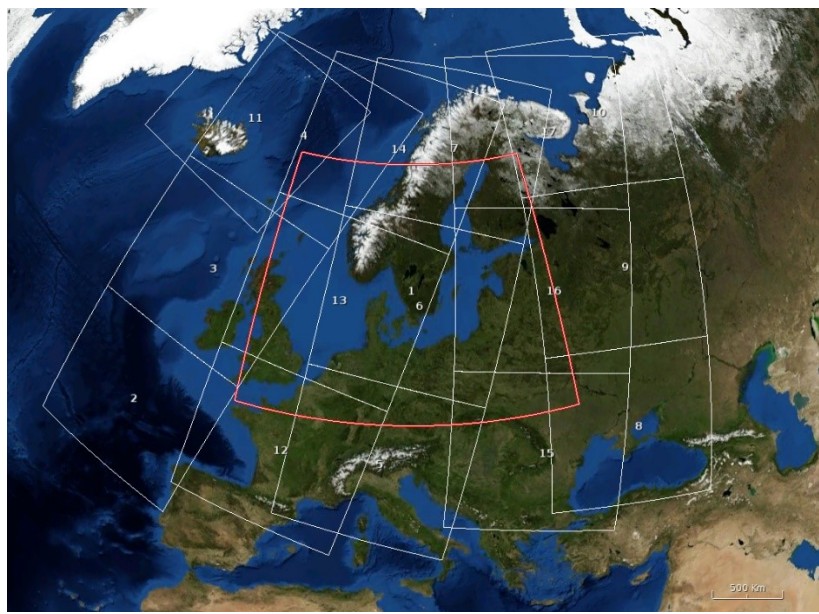


**Figure 2: Utilisation of all observational data within the region of interest from individual OLCI scenes from Sentinel-3A and 3B, which have shifted sun-synchronous orbits.**

### 2.3. Processing chain

First, a summary of the evolution of the algorithm. The used algorithm is basically a further development of the Case-2 water

algorithm by Doerffer and Schiller (2007), which was initially developed for ENVISAT MERIS and is currently being used for Sentinel-3 OLCI in the ground segment for coastal waters; this algorithm is also known as C2RCC (Brockmann et al., 2016). Parallel to the further evolution of C2RCC, an independent algorithm was branched off, the OLCI Neural Network Swarm (ONNS) algorithm by Hieronymi et al. (2017). This aims to generate reliable water quality parameters seamlessly for all natural waters and also to provide additional products, e.g. on carbon concentrations in water. The water algorithm ONNS

uses its own OWT classification with 13 defined classes and specific neural network algorithms for each class. For some time, the ONNS water algorithm (with atmospheric correction from C2RCC) was used in the Copernicus Marine Service to estimate the chlorophyll concentration in the Baltic Sea (Le Traon et al., 2021). However, the classifiability of reflectances from different atmospheric corrections (incl. C2RCC) was often insufficient, i.e. the AC methods sometimes did not provide known $R_{rs}$ shapes and some defined classes were not provided at all (Hieronymi et al., 2023a). For this reason, an independent new

AC was developed, which was optimized for the performance spectrum of ONNS and called Atmospheric Correction for Optical Water Types (A4O) (Hieronymi et al., in prep.). According to Hieronymi et al. (2023a), A4O has the following particularly good performance compared to other AC methods: 1) a high classifiability of the generated $R_{rs}$ in different OWT frameworks, i.e. basically plausible spectra in the entire spectral range (without $R_{rs} < 0$), 2) general provision of spectra in all defined classes, 3) a high spatial homogeneity of the pixel-by-pixel image analysis and thus a particularly high number of

achievable matchups. However, the comparison with in-situ measurement data shows a need for further improvement; the



reflectance spectra are often underestimated in magnitude - this is work-in-progress. Thus, the processing chain will be further optimized in terms of data quality, processing speed, data volume, and products including flags.

In the used processing chain A4O-ONNS v0.25 for daily aggregated observations, up to 17 individual OLCI Level-1 scenes of the same day with a volume of around 8 GB are processed, in the intermediate Level-2 step approx. 38 GB are produced,

and the resulting merged Level-3 NetCDF files have a size of around 3 GB. All cloud-free contributions are averaged within the superordinate grid. Due to the time offset between Sentinel-3A and 3B overflights and cloud motion, slightly more water surfaces can be seen, especially in the northern part, but this is irrelevant for the cloud statistics (Fig. 3). It should be noted, nevertheless, that the water-related products near clouds are usually subject to greater retrieval uncertainties. The optical water type classification of Bi and Hieronymi (2024) is applied to $R_{rs}$ after merging. The final product contains selected information

from Level-1 top-of-atmosphere, from climatology, the atmospheric correction, OWT classification, water products, and flags. Attention was paid to a concise metadata description, easy handling, and meaningful flagging.

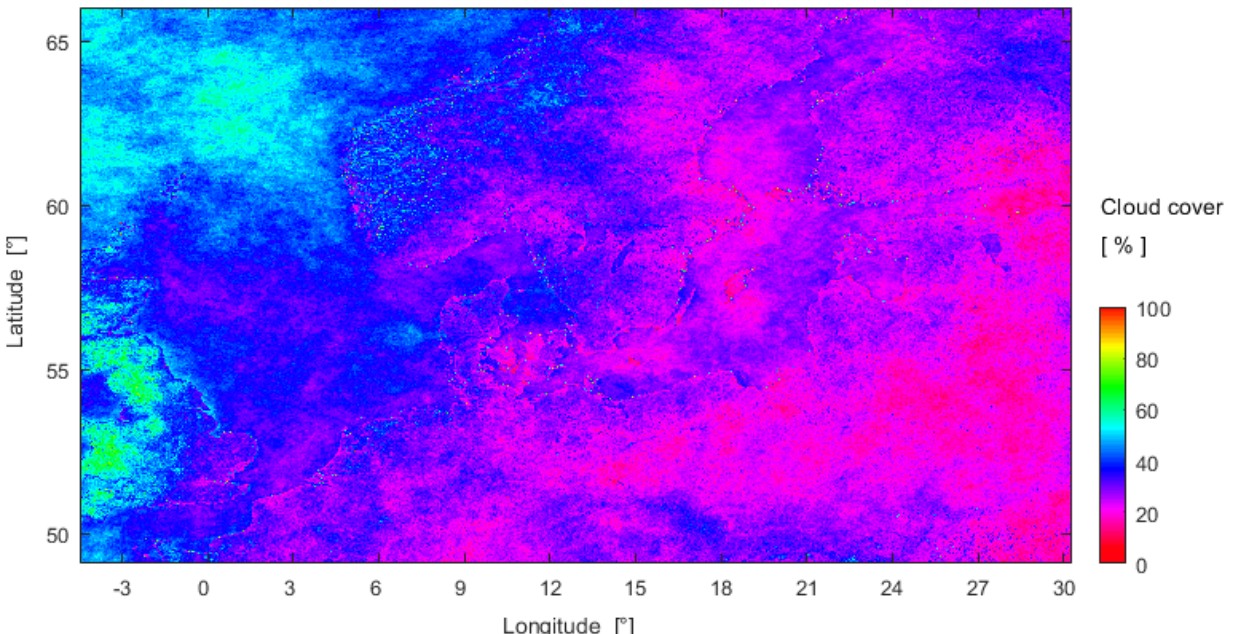

**Figure 3: Percentage cloud cover of the region over the entire period June-September 2023 (122 days) and thus indirectly clear-sky probability, which indicates pixel-based observation frequency. The coastlines can be subject to cloud misdetection.**

**2.4. Additional data**

The atmospheric correction A4O uses climatological data to better resolve influences of sea surface whitecaps and coastal conditions; these data are also provided. Climatological reanalysis data of sea surface temperature, *SST*, and salinity, *SSS*, have been adapted to A4O purposes with a global spatial resolution of 1/12° (approx. 8 km). Monthly mean temperature and yearly mean salinity were derived from the Copernicus Marine Service global ocean physics reanalysis (Lellouche et al., 2018) related

to the period 1993 to 2016. Since this product contains no data for inland waters and lakes, monthly mean *SST* of larger lakes



were derived from ERA5 data of the European Centre for Medium-Range Weather Forecasts (ECMWF, https://cds.climate.copernicus.eu/; averaged over the years 1979 to 2020 and re-gridded from 30 km resolution). Missing values for any other inland waters were filled with the mean value of all grid cells on the same latitude of each month. Grid cells without a valid salinity value represent land or freshwater and were therefore assigned to a value of zero. Major saline lakes like Dead Sea or Great Salt Lake have been included manually. In practice, this gives a temperature between 0 and 36°C and salinity values of 0 to 40 PSU. A Laplace filter is applied to the re-gridded *SST* and *SSS* during the satellite image processing to smooth grid boundaries. When using A4O, the data is projected onto the coordinates of the Level-1 OLCI scenes; in Level-3 processing, all data is projected onto the same global grid.

## 3. Presentation and context of the dataset

### 3.1. Overview of available parameters

An overview of all parameters contained in the data for each day is listed in the following, the corresponding units are in brackets, where 1 stands for unitless. Some parameters refer to the entire region of interest over land and water, such as top-of-atmosphere reflectance and wind speed. Products from climatologies and from them derived such as white cap fraction refer only to water areas resolved in the 300 m grid, i.e. for lakes, broad rivers, lagoons, and seas. Water quality characteristics are only provided for visible, cloud-free water areas. *NaN* (not a number) is used as fill values for corresponding data gaps.

Parameters from the original OLCI Level-1 data:

- L1_land: Level-1 land mask (thus implicitly water areas) [1]
- L1_Wind_speed: Omnidirectional wind speed from Level-1 [m s$^{-1}$]

Parameters from the atmospheric correction A4O (Hieronymi et al., in prep.):

- A4O_R_toa_400 – 1020: Reflectance at the top-of-atmosphere at 21 OLCI bands [1]
- A4O_Rrs_n_400 – 1020: Normalized remote-sensing reflectance at 16 OLCI bands [sr$^{-1}$]
- A4O_SST: Monthly average of the sea surface temperature from climatology [°C]
- A4O_SSS: Annual average of the sea surface salinity from climatology [1e-3] or [PSU]
- A4O_A_wc: Percentage whitecap fraction of water areas based on wind speed and sea surface temperature [1] or [%]

Parameters from the optical water type classification (Bi and Hieronymi, 2024):

- OWT_AVW: Apparent visible wavelength based on A4O_Rrs_n between 400 and 800 nm [nm]
- OWT_Area: Trapezoidal area at red, green, and blue (RGB) bands based on A4O_Rrs_n [1]
- OWT_NDI: Normalized difference index at green and red based on A4O_Rrs_n [1]
- OWT_index: Optical water type index or class based on A4O_Rrs_n [1]
- OWT_U_tot: Total membership values in the OWT framework [1]

Parameters from the ONNS water algorithm (Hieronymi et al., 2017):

- ONNS_a_g_440: Gelbstoff (CDOM) absorption coefficient at 440 nm (from IOP nets) [m$^{-1}$]
- ONNS_a_p_440: Absorption coefficient of phytoplankton particles at 440 nm [m$^{-1}$]





- ONNS_a_m_440: Absorption coefficient of minerals at 440 nm [m$^{-1}$]
- ONNS_a_tot_440: Total absorption coefficient at 440 nm of CDOM, phytoplankton, minerals, and water [m$^{-1}$]
- ONNS_b_p_440: Scattering coefficient of phytoplankton particles at 440 nm [m$^{-1}$]
- ONNS_b_m_440: Scattering coefficient of minerals at 440 nm [m$^{-1}$]
- ONNS_b_tot_440: Total scattering coefficient at 440 nm of phytoplankton, minerals, and water [m$^{-1}$]
- ONNS_a_dg_412: Absorption coefficient of detritus plus CDOM at 412 nm [m$^{-1}$]
- ONNS_b_bp_510: Total back-scattering coefficient of all particles at 510 nm [m$^{-1}$]
- ONNS_FU: Forel-Ule colour index [-]
- ONNS_K_d_490: Diffuse attenuation coefficient of downwelling irradiance at 490 nm [m$^{-1}$]
- ONNS_K_u_490: Diffuse attenuation coefficient of upwelling irradiance at 490 nm [m$^{-1}$]

Parameters for concentrations of water constituents based on ONNS-derived inherent optical properties:

- IOP_Chl: Chlorophyll concentration estimated from total particulate absorption coefficient at 440 nm [mg m$^{-3}$]
- IOP_TSM: Total suspended matter concentration estimated from total particulate scattering coefficient at 440 nm [g m$^{-3}$]
- IOP_POC: Total particulate organic carbon (POC) concentration based on inherent optical properties of phytoplankton and non-algae particles [g m$^{-3}$]
- IOP_DOC: Concentration of dissolved organic carbon (DOC) related to CDOM absorption at 440 nm (Juhls et al., 2019) [mg m$^{-3}$]

Available flags from the atmospheric correction and water algorithm

- A4O_flag_cloud: Cloud mask from A4O [1]
- A4O_flag_cloud_risk: High risk of clouds, which can strongly influence the quality of the retrieval [1]
- A4O_flag_adjacency: Pixel near land or clouds with high risk of retrieval influence, e.g. through sub-pixel contamination of land, optically shallow water, aquatic plants or cloud edges [1]
- A4O_flag_bright: Level-2 bright mask, that includes possible clouds and sea ice [1]
- A4O_flag_suspect_pixel: Level-2 flag for possibly implausible AC output [1]
- A4O_flag_floating: Level-2 flag for very high biomass and possibly floating algae [1]
- A4O_flag_glint_risk: Level-2 flag for sun glint risk [1]
- ONNS_limited_valid: Estimated IOPs and concentration values are highly uncertain [1]

Additional information on Level-3 data aggregation

- TOA_count: Number of pixels from available satellite images at the top-of-atmosphere    [1]
- BOA_count: Number of pixels from available cloud-free images after atmospheric correction at the bottom-of-atmosphere [1]

Background information on selected parameters is provided in the following.

## 3.2. Whitecap fraction

Information on the whitecap fraction is important, for example, for estimating the gas exchange at the air-sea interface, the heat flux, or the formation of marine aerosols. Despite the relatively small area of coverage, whitecaps are relevant for atmospheric correction due to their high reflectivity in contrast to dark water. The surface fraction covered with whitecaps, $A_{wc}$, depends on wind speed and water temperature and is usually much smaller than 5 %. Whitecap fraction is parameterized based on global microwave satellite observations (with a frequency of 10 GHz; Albert et al., 2016). The estimated $A_{wc}$ is



provided by the atmospheric correction A4O together with wind speed from OLCI Level-1 and sea surface temperature from monthly climatology, *SST*.

### 3.3. Concentrations of water constituents

The original ONNS algorithm by Hieronymi et al. (2017) applied an internal OWT classification and class-specific neural network (NN) algorithms to derive IOPs, light field parameters, and concentrations of water constituents directly. One reason

for this was internal checks of system uncertainties. For a clearer, more flexible and purely physics-based derivation of the water constituents, the concentrations are now determined based on the NN-derived IOPs only. This is essentially the approach already favoured by Doerffer and Schiller (2007) and used in the OLCI Level-2 processing for Case-2 waters. To emphasise this, the concentration labels refer to IOPs and not to ONNS directly. Moreover, for concentrations of chlorophyll and total suspended matter, similar IOP-relationships were used as in Doerffer and Schiller (2007).

The chlorophyll concentration is linked to phytoplankton-pigment absorption, but due to previous modelling inadequacies in the optical component separation, it is estimated here from total particulate absorption at 440 nm (an interim approach that will be revised in future):

$$\text{IOP\_Chl} = 21 \, (a_p + a_m)^{1.04}. \tag{1}$$

The concentration of total suspended matter is estimated from total particulate scattering coefficient at 440 nm with:

$$\text{IOP\_TSM} = 1.73 \, (b_p + b_m). \tag{2}$$

The analysis of optical measurement data (Röttgers et al., 2023) and concentration of particulate organic carbon, *POC*, shows clear dependencies on phytoplankton and mineral particles or detritus, especially for coastal waters. In addition to phytoplankton absorption at 440 nm, ONNS also derives the total backscattering coefficient of all particles at 550 nm. The concentration of particulate organic carbon is well represented by the following IOP-relationship:

$$\text{IOP\_POC} = 5.5 \, b_{bp}(510) + 2.7 \, a_p(440). \tag{3}$$

Juhls et al. (2019) show that the concentration of dissolved organic carbon, *DOC*, can be estimated via the absorption coefficient of coloured dissolved organic matter at 440 nm (in the ONNS notation, subscript *g* stands for *Gelbstoff* as a synonym for *CDOM*). Their relationship is also applied here:

$$\text{IOP\_DOC} = 10^{2.525} * a_g^{0.659}. \tag{4}$$

### 3.4. Parameters from the Optical Water Type classification

Hieronymi et al. (2023a) documented weaknesses of previous OWT classification methods, including the one implemented in the ONNS algorithm. To mitigate these shortcomings, Bi and Hieronymi (2024) presented a new holistic OWT classification for ocean, coastal, and inland waters, the results of which are included in the dataset. The starting point for the classification is spectral remote-sensing reflectance provided by the atmospheric correction A4O. However, unlike previous OWT

frameworks, $R_{rs}$ is not compared directly with class-mean spectra, but three optical variables are derived from $R_{rs}$, which are compared in their covariances. The spectral magnitude is described by *OWT_Area*, the trapezoidal area below the $R_{rs}$ spectrum



at three red, green, and blue (RGB) bands, also referred to as the zeroth spectral moment. The apparent visible wavelength, *OWT_AVW*, describes the weighted average of the wavelengths between 400 and 800 nm, which can be interpreted as the centroid of the spectrum, also described as the first spectral moment. The normalised difference index between green and red,

*OWT_NDI*, helps to distinguish between phytoplankton and detritus-dominated spectra. Bi and Hieronymi (2024) define ten different water types, whereby OWT labels with the endings "a" and "b" were intended for types with a similar spectral shape but differing magnitudes, with "b" usually representing a variant with higher brightness. During the OWT analysis of the reflectance, weights are assigned to the defined classes, with three to six classes usually contributing. Thus, in addition to the index for the water class with maximum membership, the total membership of all contributing classes is also provided in the

dataset (*OWT_index* and *OWT_U_tot*). The total membership serves as an indicator of the quality of the classifiability; a minimum requirement of 0.0001 is often used (e.g. Hieronymi et al., 2023a).

Figures 4 to 8 illustrate the available data for one day (July 8, 2023) of the merged satellite image at the top-of-atmosphere, after atmospheric correction, as well as the three optical variables for the OWT classification. Figure 9 shows the resulting OWT classes with the highest membership for that day. In most cases, one class dominates in the OWT analysis and intuitive

conclusions can be drawn about the reflectance spectrum (and corresponding water constituents or phytoplankton groups). At the boundaries between dominant water types, there is often a transition zone with approximately equal contributions from neighbouring types. For a full exploitation of the OWT method, the results of the specific water algorithms are mixed with corresponding weights in a fuzzy-logic approach.


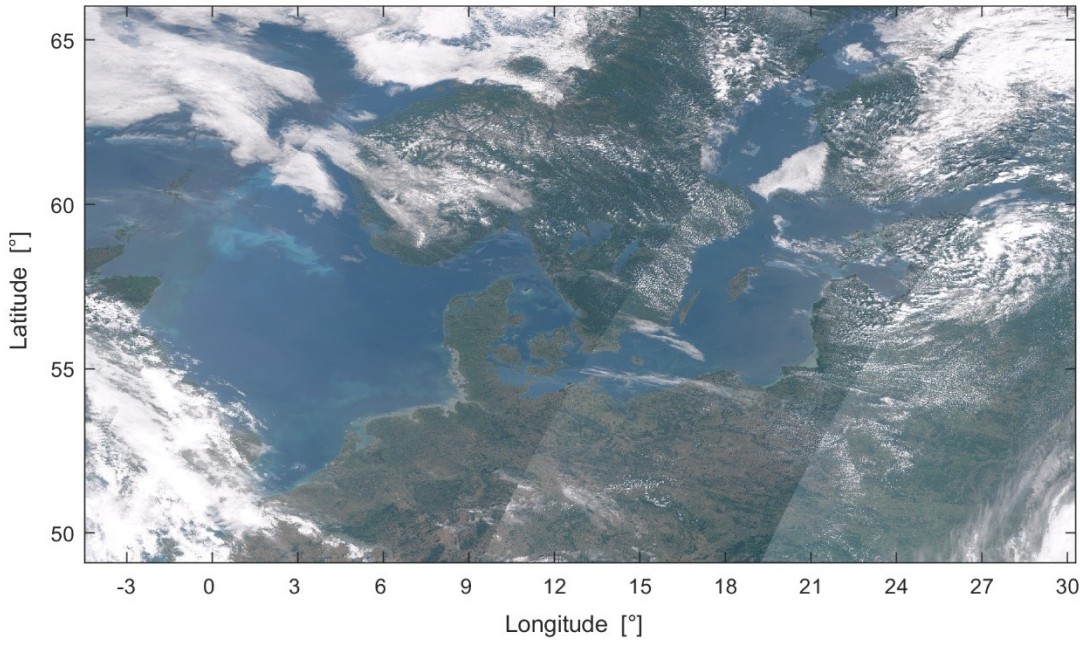

**Figure 4: RGB image created from reflectance top-of-atmosphere for one day (July 8, 2023). It illustrates the starting point for the**
**atmospheric correction including clouds, water-land areas, and observation angle effects at the original image margins. TOA information is also used for orientation over land and in case of cloud issues.**

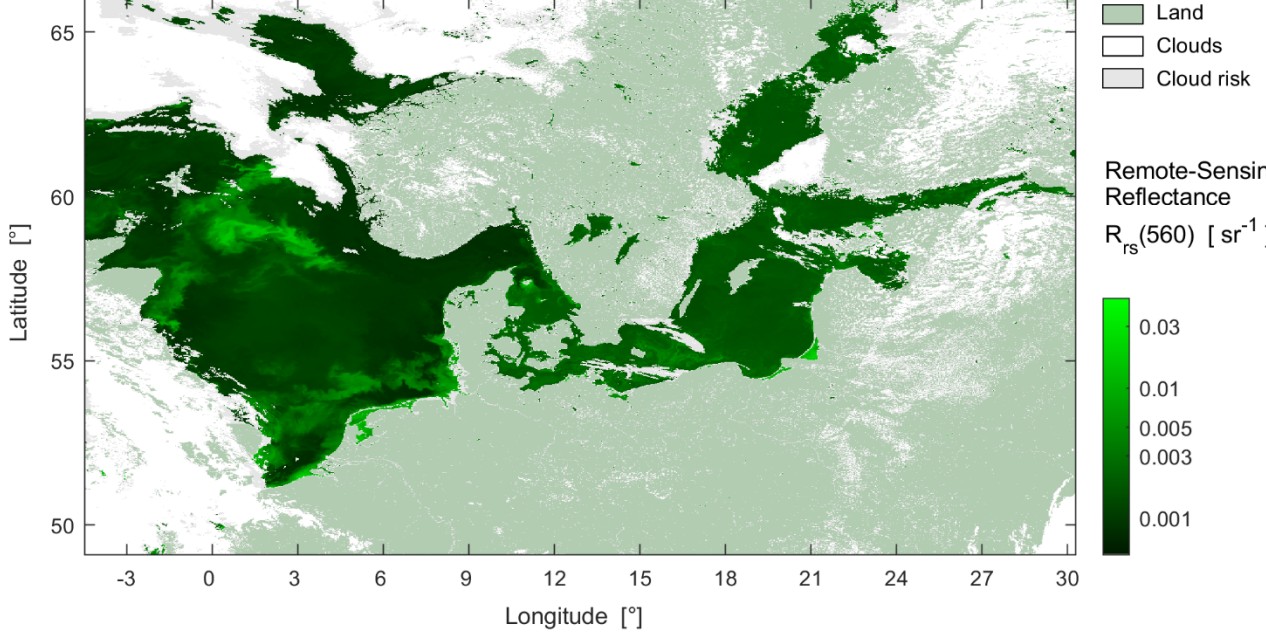

**Figure 5: Corresponding to Fig. 4, the result of the atmospheric correction: normalized remote-sensing reflectance at one wavelength (the green band at 560 nm) and labelling of land, clouds and cloud risk. Original image boundaries with increased "air mass" are**
**no longer recognisable, only at cloud discontinuities.**

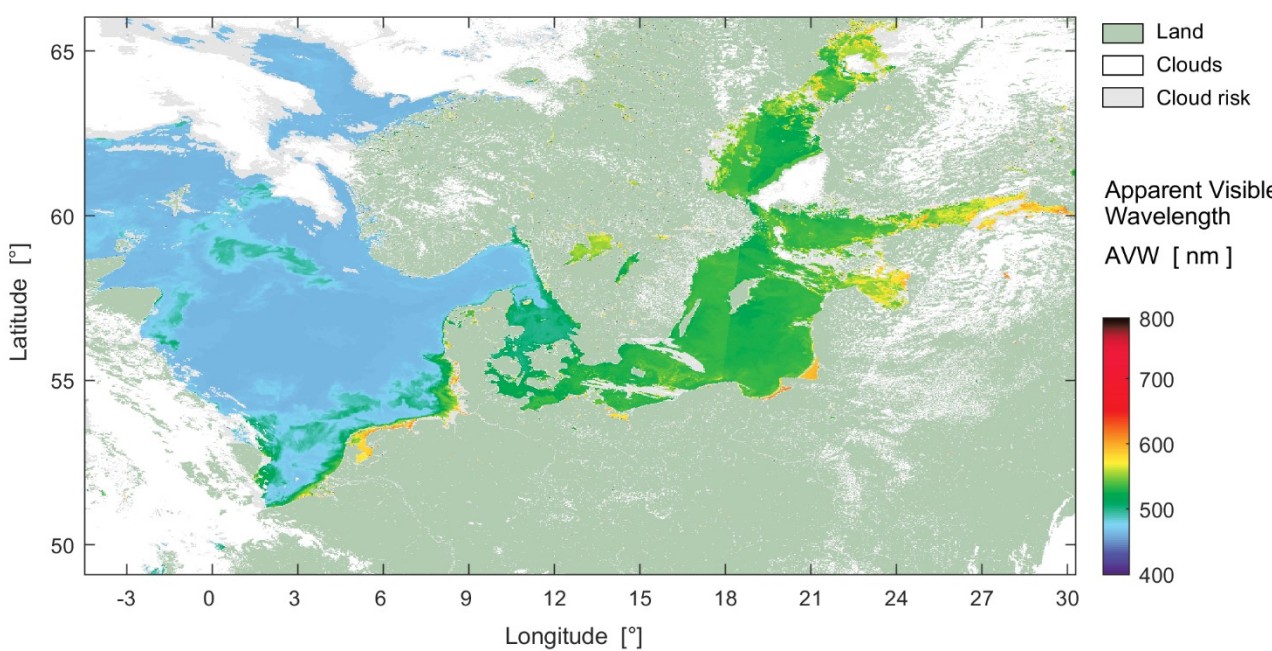

**Figure 6: Corresponding to Fig. 5, derived apparent visible wavelength with reference to the spectral range 400 to 800 nm for OWT classification.**


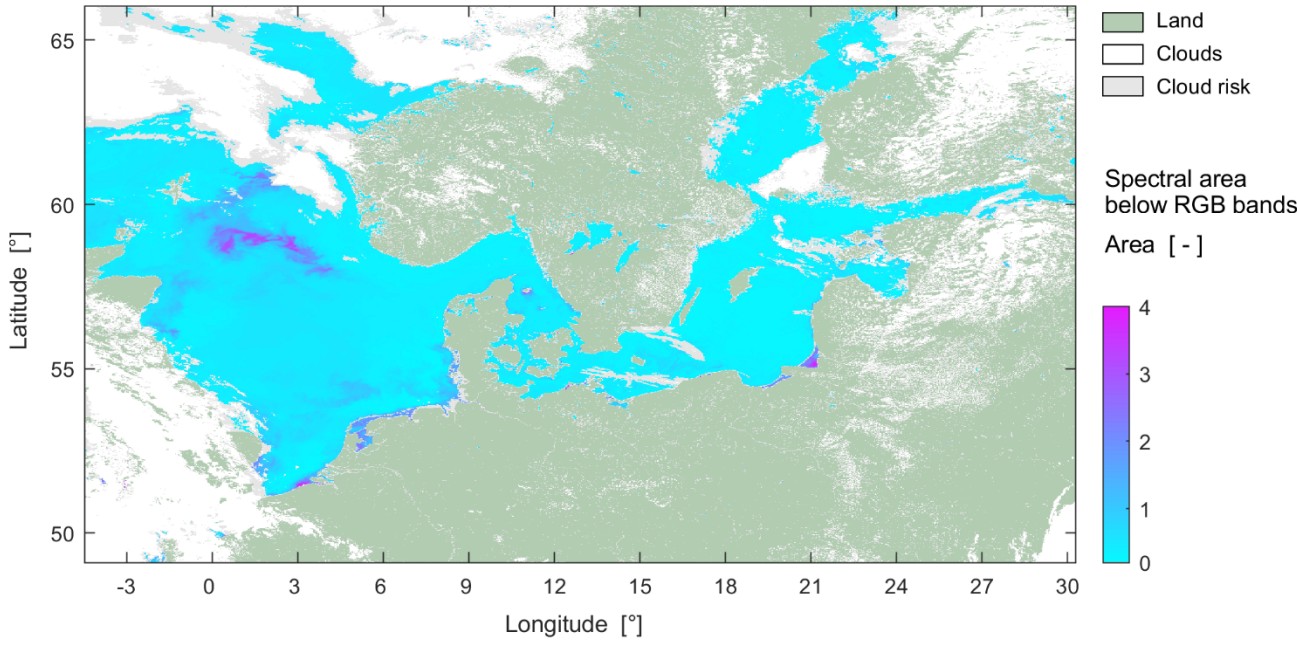

**Figure 7: Corresponding to Fig. 5, derived spectral area below $R_{rs}$ at 443, 560, and 665 nm for OWT classification.**

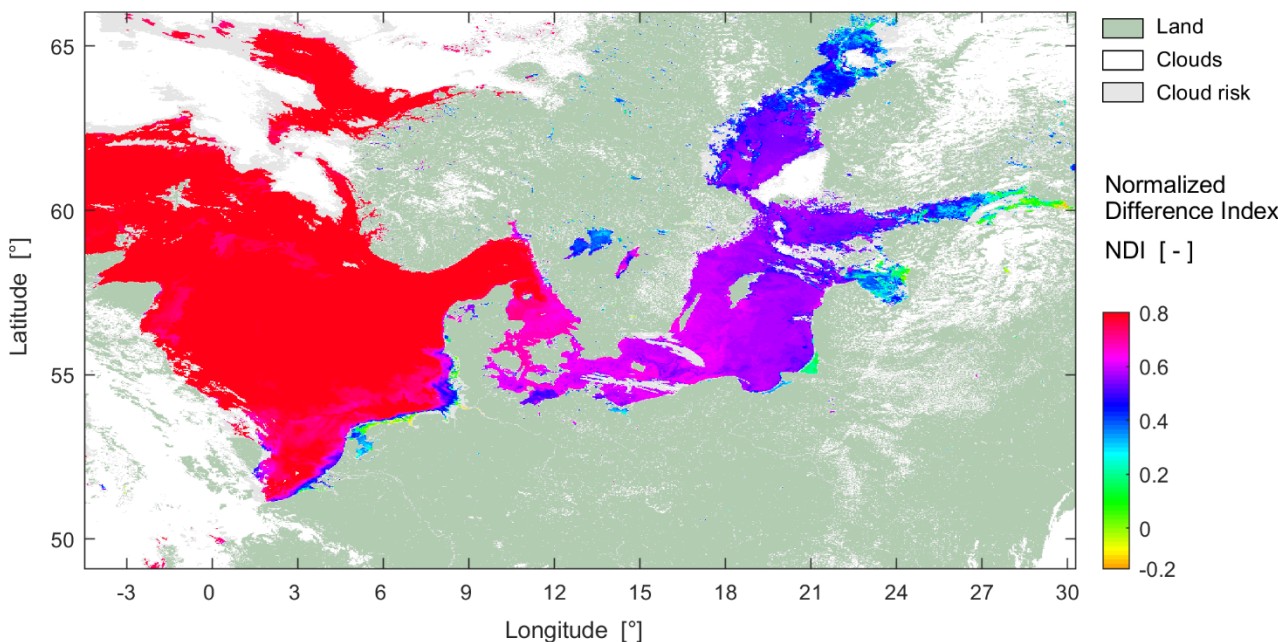

**Figure 8: Corresponding to Fig. 5, derived normalized difference index for OWT classification.**

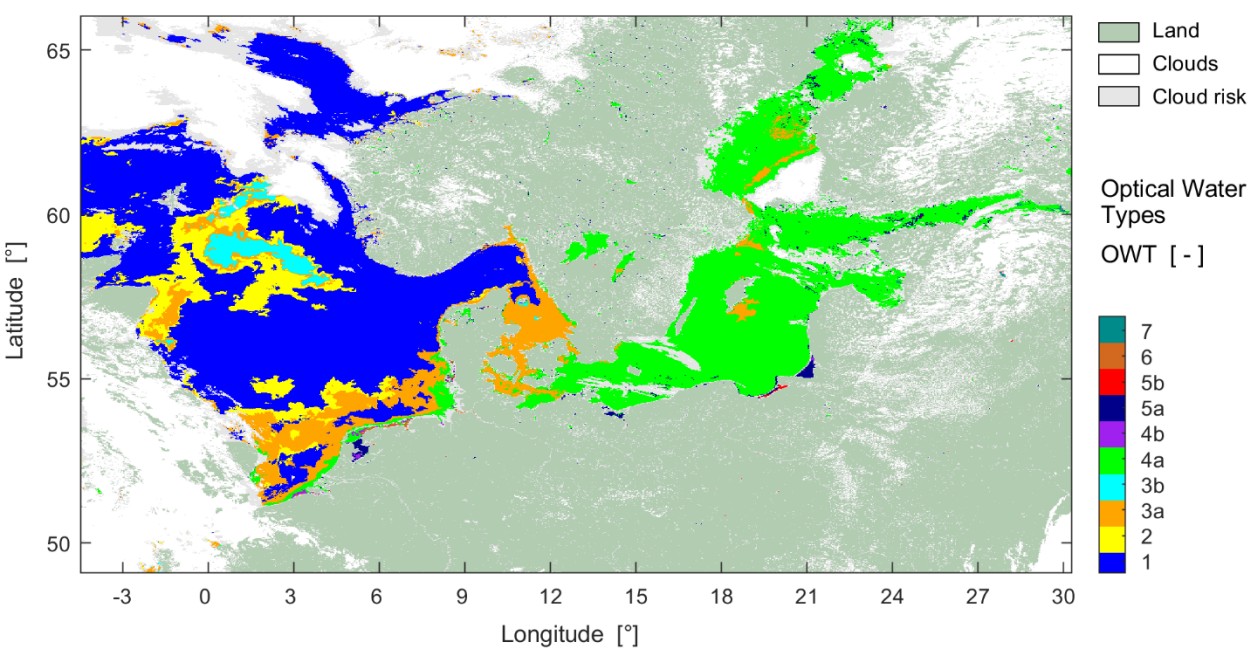


**Figure 9: Corresponding to Fig. 5, resulting water types with maximum membership. Seven spectral $R_{rs}$ shapes are distinguished in the OWT framework; the subdivision into "a" and "b" shows differences in magnitude.**



## 4. Assessment of the optical complexity of the water bodies

This section demonstrates the application potential of the optical water type classification data through a representative analysis of summer 2023 patterns in the North Sea and Baltic Sea region, showcasing how the dataset can elucidate water mass dynamics and ecological transitions. Figure 10 shows a map of the water classes with maximum memberships that occur most frequently in the four-month period. This demonstrates a fundamental distinction between North Sea and Baltic Sea, which also justifies the regional split of the products in the Copernicus Marine Service (Brando et al., 2024). In coastal and inland waters, the variability is broader with possible problems of the water algorithms used. Hieronymi et al. (2023a) compared five AC methods for OLCI and showed that especially in the transition from coastal to clearer North Sea waters, an ambiguous variability of $R_{rs}$ shapes is provided by the different AC methods, with A4O having a possibly erroneous tendency towards more blue water (i.e. OWT 1). The other AC methods would probably result in larger areas with OWT 2 to 4-like reflectances. However, Hieronymi et al. (2023a) also documented some significant problems with the plausibility and classifiability of the results from the other atmospheric correction methods, e.g. to large-scale overcorrection with (incorrect) $R_{rs} < 0$ or significant excess of $R_{rs}$ especially in blue bands. This requires further studies and comparisons with in-situ measurements.

In general, however, the regional distributions show features that are reasonable from an oceanographic-limnological point of view. Figure 9 shows areas in the central North Sea with OWT 3b, a class that is intended to represent bright-turquoise blooms of coccolithophores, which contain high amounts of particulate inorganic carbon (calcite particles). Such bright pixels are sometimes flagged as clouds in other algorithms; in any case, such cases are critical regarding the assumptions for chlorophyll estimation, but OWT-specific assumptions can help here. But the coccolithophore bloom (OWT 3b) in Fig. 9 is only a temporary event with a phenologically changing colour appearance like shown for example in Cazzaniga et al. (2021); blue water (OWT 1) is the actual background in that region during summertime (Fig. 10). Another feature is visible in the North Sea area at the Dogger Bank with OWT 2, where the water can be less than 20 m deep, which on the one hand can lead to hydrodynamic resuspension of sediments through waves, and on the other hand it is possible that influences of the shallow bottom can be seen. This shows a fundamental problem of ocean colour, namely that optically shallow waters cannot yet be reliably flagged without water depth information, which can lead to misinterpretations of biomass and other parameters. The problem of shallow water and visible seafloor is particularly visible at the island of Læsø in the Kattegat, where the OWT variability is very high - in fact, this should be flagged, because underlying assumptions for the water algorithms are invalid. But even apart from such artefacts, the map shows that all defined OWTs occur, and the region thus well represents the optical variability of natural water bodies. But again, it is possible that A4O estimates something too "blue spectra" in relatively clear shelf sea water - however, there are also significant contributions from OWTs 2 and 3a.

Also of interest is a map showing the number of different classes with maximum membership over the entire period (Fig. 11). Typical water algorithms and atmospheric corrections may be out-of-scope for different OWTs, which is not the case for A4O and ONNS. Areas with many different OWTs indicate high dynamics of sediments or algal blooms with possible eutrophication issues. Examples of water bodies with high optical variability with up to six different dominant classes are some





Estonian lakes, including the large Lake Peipus. Ansper-Toomsalu et al. (2024) compared satellite products with in-situ data from these waters, including the A4O-ONNS processing described here. Their results show a need for improvements but are
not yet conclusive. Ideally, the comparisons should be made in the context of the prevailing OWT, which can help the further development of the algorithm and formulation of uncertainties. In principle, there can also be differences in the reflectance shape and the allocation of the OWT due to clouds, subvisible clouds, cloud shadows, but also due to subpixel contamination by coastal vegetation such as reeds, which is then interpreted as high biomass class. Nevertheless, there are also areas that are optically dominated by only one class over the entire period, such as the tide-influenced muddy and very bright Bristol Channel
with dominating OWT 6. For ground-truthing and definition of system vicarious calibration gains, low optical variability over time is more important (but rather for clear waters). Analysing the recurring patchiness of water classes can support the selection of representative sites for water quality monitoring and satellite validation (Lehmann et al., 2021).

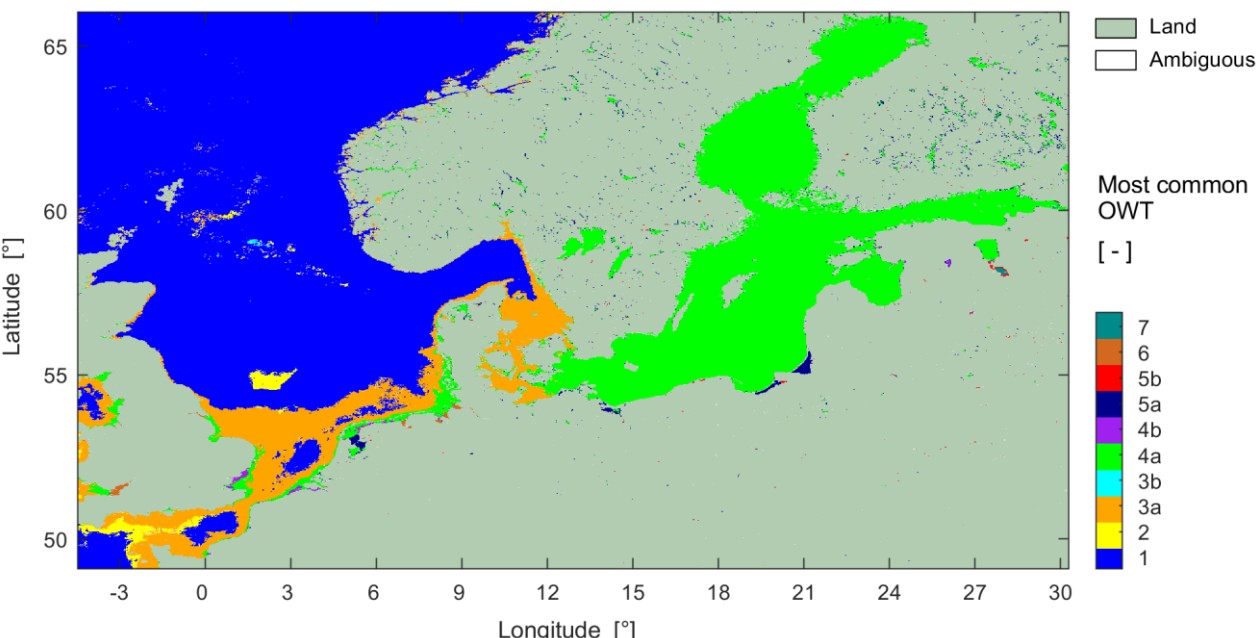

**Figure 10: Most common water types with maximum membership within June-September 2023.**





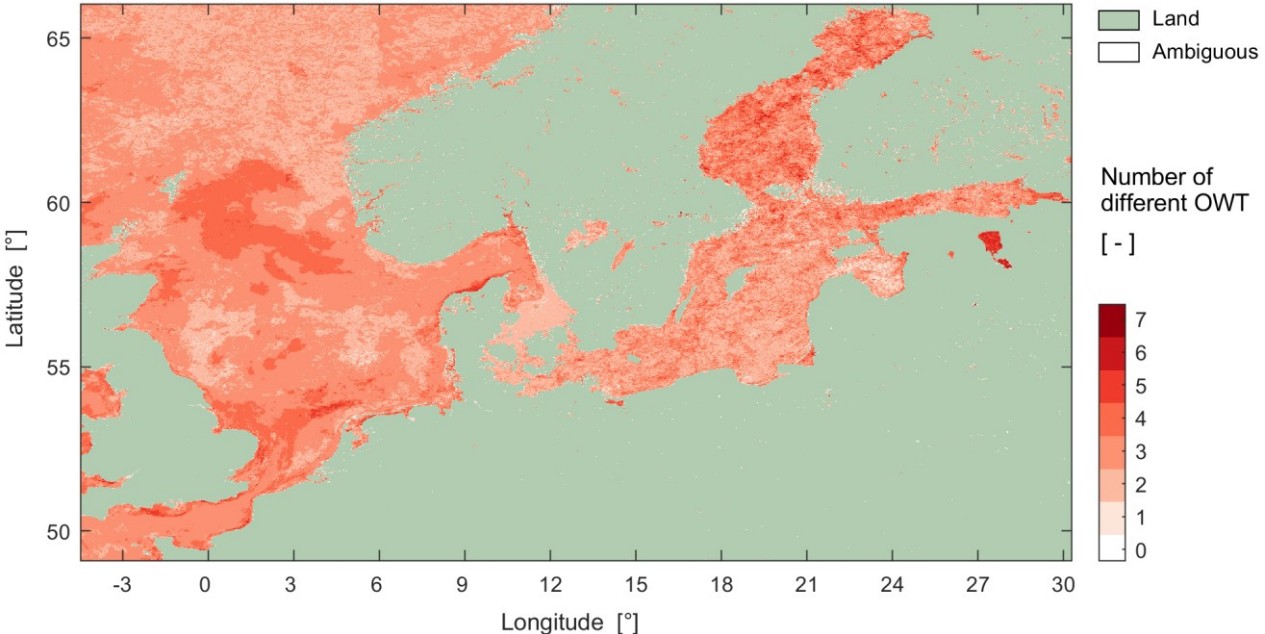


**Figure 11: Number of different water types with maximum membership within June-September 2023, i.e. the ecologically and hydrodynamically caused changes in water colour during the summer months.**

The occurrence of water classes and their classifiability for the entire region and period is shown in Fig. 12. This is a form of visualisation as already used in Hieronymi et al. (2023a) and Bi and Hieronymi (2024) (and their supplement), and in

comparison with their results, Fig. 12 shows that $R_{rs}$ from A4O can be well-classified with this OWT framework (99.68% of the cases lead to a minimum total membership of 0.0001) and that all classes are filled. This is a significant benefit of A4O, but also of the OWT framework.

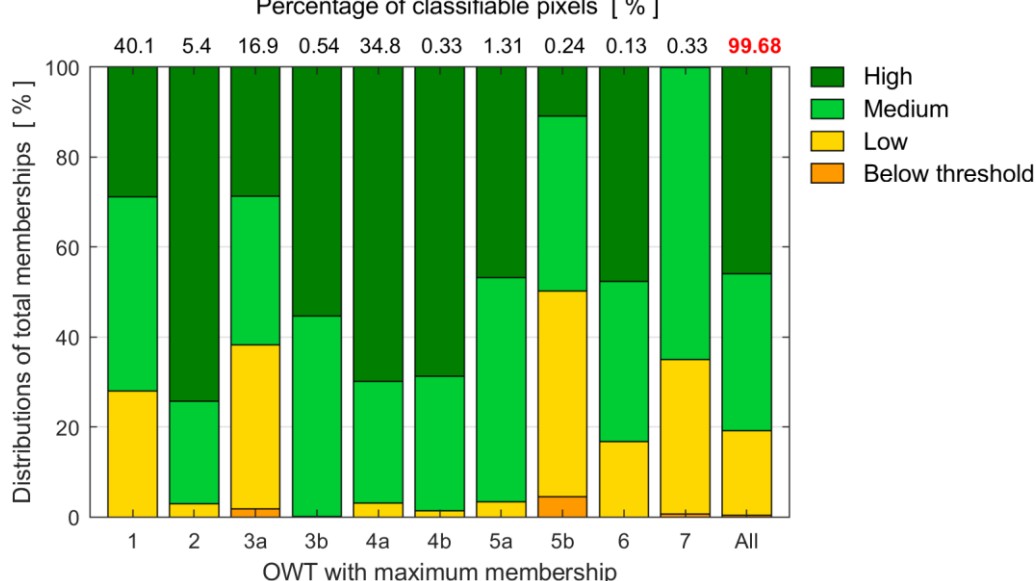

**Figure 12: Distribution of optical water types with maximum membership and category of their total memberships in high (>0.8), medium (0.3-0.8), low (0.0001-0.3), and below threshold (0-0.0001) for the entire region and period. The bar on the right gives the overall distribution. The figure corresponds to the illustrations in Hieronymi et al. (2023a) and Bi and Hieronymi (2024).**

Using the regional subdivisions shown in Fig. 1, which are aligned with the domains in the Copernicus services, the optical properties of the water areas can be further specified. Table 1 shows the distribution of occurring water classes with maximum memberships and the overall level of classifiability in the subdomains. The very few non-classifiable cases, where the sum of the class memberships does not exceed the threshold value of 0.0001, are mostly associated with clouds, e.g. in 0.53% of cases for the North Sea where the overall cloud cover is also higher (Fig. 3).

In the North Sea region (which here includes other parts of the European North-West Shelf Sea), the oceanic water types 1 to 3a dominate (with a total of 96.2%). This would also be expected from other AC methods (not shown). The bloom of coccolithophores visible in Fig. 9 is characterised by OWT 2, 3a, and 3b. The transitional waters to the southern coasts are more strongly influenced by CDOM input and sediment resuspension (OWT 4a and 4b). Thus, the spatial distribution of the classes also reflects the bathymetry of the North Sea.

The Baltic Sea is optically dominated by CDOM absorption effects and OWT 4a (81.3%). The Skagerrak-Kattegat region is the transition to clear waters and marine salinity. As mentioned, there are some cases of optically shallow water with a visible sea floor for which OWT classes can serve as a mask, because the usual model assumptions for estimation of water constituents do not apply. OWT 5a and 5b also occur in the Baltic Sea. These classes represent green eutrophic to hypereutrophic waters with significantly higher phytoplankton biomass and bimodal reflectance shape. In the Baltic Sea, this is mostly associated with cyanobacteria blooms. The occurrence of these water classes can therefore be used directly as an indicator for cyanobacterial blooms. OWT 7 stands for dark brown waters with a very high CDOM concentration and low reflection in the



entire visible range. This class of water is found in the Baltic Sea particularly near river outflows in the Gulf of Finland and the Gulf of Bothnia.

Inland waters are generally more characterised by CDOM absorption and its reflectance attenuation in the blue. Accordingly, oceanic water classes are not listed as predominant in this region. However, this could be the case in other regions of the world, e.g. in very clear oligotrophic lakes. Most inland waters fall into classes 4a, 5a, and 7, all of which have relatively high CDOM effects. Over 30% of the water bodies also have high concentrations of algae (5a and 5b), therefore the OWTs are also characteristic for the trophic state of inland waters. OWT 6 stands for extremely scattering (NAP-dominated) turbid waters; they occur in some river estuaries like the tide-influenced Lower Elbe or Severn that flows into the Bristol Channel.

The category of coastal waters consists of parts of the other three subdomains and is optically the most diverse. Coastal waters range from oligotrophic ocean to hyper-eutrophic algal blooms and from bright-scattering to dark-absorbing. This means that the use of the term "coastal" for ocean colour algorithms is in fact somewhat misleading. The OLCI Level-2 coastal algorithm (or more precisely for optically complex waters), i.e. C2RCC, is optimised for moderately scattering or absorbing waters (Hieronymi et al., 2023a).

**Table 1: Percentage distribution of occurring water classes in regional subdomains and overall classification level.**

| OWT | 1 | 2 | 3a | 3b | 4a | 4b | 5a | 5b | 6 | 7 | All |
|---|---|---|---|---|---|---|---|---|---|---|---|
| North Sea | 66.0 | 9.0 | 21.2 | 0.9 | 2.3 | 0.29 | 0.07 | 0.03 | 0.14 | 0 | 99.47 |
| Baltic Sea | 3.6 | 0.36 | 11.1 | 0.03 | 81.3 | 0.33 | 2.2 | 0.32 | 0.1 | 0.64 | 99.98 |
| Inland waters | 0 | 0 | 1.16 | 0 | 61.9 | 1.74 | 24.2 | 6.0 | 0.55 | 4.4 | 100 |
| Coastal waters | 18.1 | 3.1 | 20.3 | 0.13 | 54.4 | 0.72 | 2.14 | 0.33 | 0.29 | 0.47 | 99.86 |
| Total | 40.1 | 5.4 | 16.9 | 0.54 | 34.8 | 0.33 | 1.31 | 0.24 | 0.13 | 0.33 | 99.68 |

## 5. Other features of the dataset

The dataset also contains several parameters that are useful for oceanographic-limnological, atmospheric, and biogeochemical process studies. Examples are outlined in the following.

### 5.1. Clouds

From global MODIS cloud observations, it is estimated that the fraction of clouds over land is about 55%, with a distinctive seasonal cycle, whereas cloud cover over the oceans is around 72%, with reduced seasonal variability (King et al., 2013). In our region of interest, summer tends to have fewer clouds than winter, and our summer-dataset exhibits an overall lower cloud fraction compared to global climatology. The cloud cover averaged over the North Sea, Baltic Sea, and land areas (including inland waters) shows a relatively large daily variability and is significantly higher for the North Sea (41%) than for the Baltic



Sea and land areas in the region (both approx. 27%) (Fig. 3). At the British Isles and north of them, cloud cover was on average

often >60%. The coastlines, and here in particular bright sandy beaches, are often misinterpreted as clouds (or cloud risk), which can be used for better flagging in the future. The cloud cover has an impact on the observable areas and possible matchups with satellites. Cloud cover also has an influence on the available solar radiation for photosynthesis of phytoplankton and is therefore crucial for estimating primary production. For example, neglecting photoacclimation of phytoplankton under clouds likely leads to a significant underestimation of growth rate and therefore of primary production (Begouen Demeaux et

al., 2025). The available reflectances at top-of-atmosphere can be used to determine cloud shapes and brightness for cloud statistics, but also for better cloud-flagging and cloud-shadow detection in the future. Cloud motions and different correction methods also create problems when merging ocean colour data from different satellite missions for long time series (van Oostende et al., 2022).

### 5.2. Wind

The wind speed, i.e. the omnidirectional horizontal wind vector at 10 m altitude, is transferred from the OLCI Level-1 data and is available for all land-sea areas. Above water, wind influences the roughness of the water surface and wave development, the size of the direct sunglint area, the fraction of whitecaps at the surface due to wave breaking, the initial diffuseness of the underwater light field, but also the mixing of the upper water layer (e.g. Hieronymi and Macke, 2012; Hieronymi, 2016). This means that wind has an influence on the performance of the atmospheric correction, which is usually designed for moderate

wind speeds, which is guaranteed on average for the North Sea and Baltic Sea at 6.3 and 5.5 m s$^{-1}$ respectively. However, there were also large-scale storm events during the observation period, e.g. at the beginning of August, which brought large amounts of precipitation to Scandinavia. Strong wind events also lead to surface-accumulated algae being mixed into the depths.

### 5.3. Carbon in the upper water column

Brewin et al. (2023) provide an overview of methods for obtaining ocean carbon from space, but also priorities, challenges,

gaps, and opportunities for satellite estimates. Their review targets on both inorganic and organic pools of carbon in the ocean, in both dissolved and particulate form, as well as major fluxes of carbon between reservoirs (e.g. primary production) and at interfaces (e.g. air-sea and land-ocean). Extreme events, "blue carbon", and carbon budgeting were also key topics discussed. Current algorithms include those that are: based on empirical band-ratio or band-differences in remote-sensing reflectance wavelengths; IOP-based; IOP and chlorophyll based; based on estimates of diffuse attenuation ($Kd$); and based on relationship

between diffuse attenuation and IOPs. Our dataset contains most needed parameters required for the use of the various algorithms. Some information can be estimated from the local course of the sun and large-scale cloudiness, like the diurnal photosynthetically available radiation (PAR) that is essential for estimating primary production. Optical water types also serve the better characterization of aquatic carbon. OWT 3b for example, which is especially defined for strongly backscattering coccolithophore blooms, serves as a direct hint for particulate inorganic carbon (PIC). The IOPs and reflectances provided

allow quantification of PIC concentrations (e.g. Balch and Mitchel, 2023). The flag for floating algae is an indicator that



recognizes scum at the surface or floating macroalgae such as *Sargassum*, which are part of the "blue carbon". There are therefore extensive links in the dataset for estimating aquatic carbon.

We pursue the physics-based approach of determining the inherent optical properties of individual water constituents from aquatic reflectance. The optical effect of biogeochemical stocks can be reliably estimated using IOP-constituent relationships.
The establishment of such relationships increases the traceability of individual processing steps. The concentration products in the dataset are based exclusively on the estimated IOPs and do not yet include any system vicarious calibration for the overall processing, which is an option for optimising validation results (e.g. O'Kane et al., 2024). In addition to the estimation of widely used concentrations of chlorophyll and total suspended matter, we also provide concentrations of dissolved and particulate organic carbon (*DOC* and *POC*). However, a matchup-based and OWT-specific validation of the products is still
ongoing. Nevertheless, our *POC* measurements in the North Sea region usually show values of 0.05 to 0.5 g m$^{-3}$; in the turbid parts of the German Bight and in the Elbe estuary, *POC* values up to 15 g m$^{-3}$ have been measured (Röttgers et al., 2023). Our unpublished measurement data from the central Baltic Sea within the time frame of the dataset (July 2023) show *POC* between 0.5 and 5 and some outliers up to about 10 g m$^{-3}$ (Hieronymi et al., 2023b). These are exactly the magnitudes of *POC* concentrations that are mirrored in the dataset and shown in the monthly average in Fig. 13. Existing *POC* algorithms are often
optimised for lower concentration ranges of oceans, e.g. Stramski et al. (2008), but there are also others that target higher coastal concentrations such as Loisel et al. (2023). Ultimately, their performance also depends on accurate atmospheric correction. The aim of our efforts with A4O and OWT-specific IOP-concentration relationships is to cover the entire span of concentrations over several orders of magnitude.

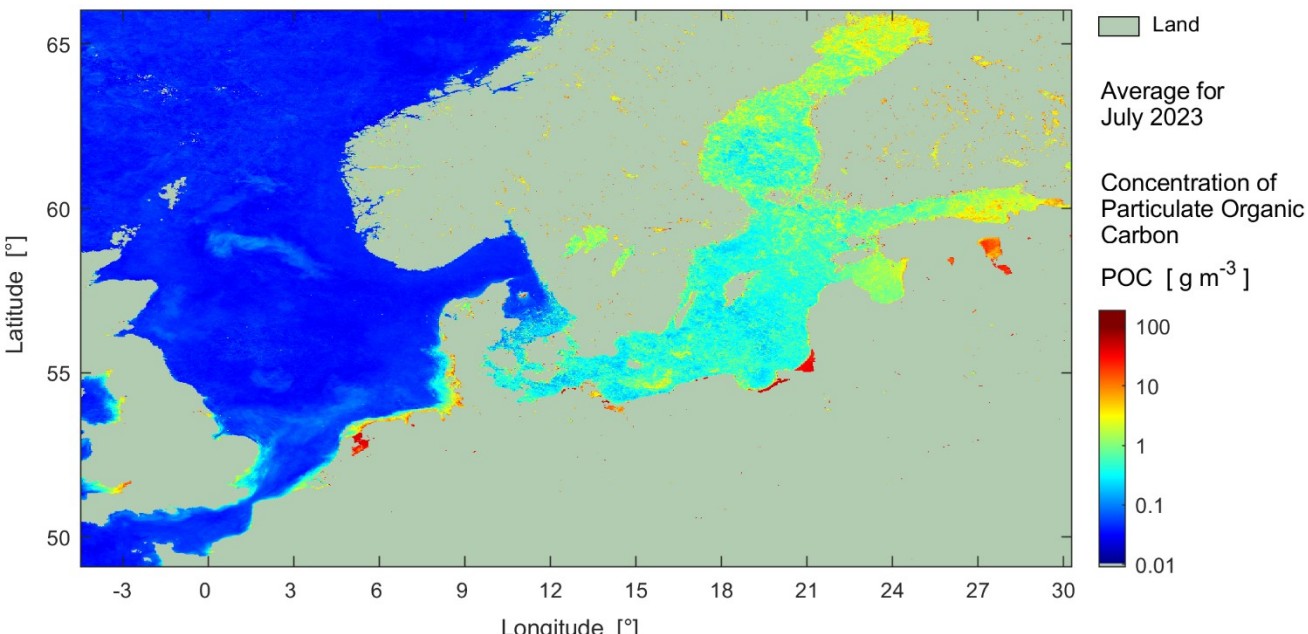

**Figure 13: July-monthly-averaged IOP-based estimate of the concentration of particulate organic carbon in the upper water column.**



## 6. Uncertainties of the dataset

The complete satellite image processing chain of atmospheric correction, OWT results, IOP algorithm, IOP-constituent relationships, and day-aggregated data is still work in progress and requires better harmonisation of the components. Individual parts, such as the atmospheric correction (A4O) and the water algorithm (ONNS), provide each an estimate of the uncertainties for the derived outputs – but these sub-results are not provided in the present dataset. These original uncertainty estimates are sometimes misleading because they do not take into account the other components, especially the coupling of atmospheric correction and water retrieval, but also not the higher-level flagging and actual comparisons with in-situ measurements.

The spatial visualisation of the data gives indications of occasional imperfections, e.g. regarding flagging of clouds, nearby land, shallow water, sun glint, and tidal flats, but also viewing geometry effects from the atmospheric correction, which are sometimes mirrored in water products. Parameters such as CDOM absorption coefficient (and the corresponding directly derived *DOC* concentration) are particularly sensitive to smallest errors of the atmospheric correction due to the ambiguities of the system and the absolute signal dominance of Rayleigh light scattering in the atmosphere. Very high concentrations of CDOM are usually found in small inland waters and are discharged into the sea via rivers, which can lead to additional risks in monitoring due to relatively coarse spatial resolution and land-adjacency effects (e.g. El Kassar et al., 2023).

The ambitions of the overall algorithm are very challenging, especially since a biogeo-optical system is to be described in an all-water type-comprehensive and open-connectable way, but all parameters basically vary over several orders of magnitude in their value ranges. Mission requirements for uncertainties are often defined in a simplified manner in the context of Case-1 and Case-2 waters, but requirements are not specified for all variables, e.g. IOPs (e.g. IOCCG, 2019). A more concrete OWT-specific guideline could help here; however, this also includes extreme waters that can exhibit very high uncertainties.

The underlying algorithm setup has not yet been sufficiently validated for each water type; some variables and preliminary elements were addressed in Hieronymi et al. (2023a) and Ansper-Toomsalu et al. (2024). On the one hand, there are many parameters that are useful for the spectral system description and serve as links for various empirical algorithms; on the other hand, some parameters are rarely measured and are not available for all defined water types. In fact, the current efforts of FAIR data preparation are aimed at finding suitable validation data and contextualising the satellite products. The user-oriented, traceable, and simple categorization of uncertainties is a focus of ongoing research.

## 7. Outlook

An obvious and important goal is the appropriate validation of the parameters of the dataset and the processing chain in general, especially those related to water quality. Methodologically, five groups with specific requirements are distinguished here: remote-sensing reflectance (16 values), diffuse attenuation of irradiance (two), colour index (one), inherent optical properties (nine), and concentrations of water constituents (four). We are convinced that it would be beneficial to carry out the validation in the context of optical water type classification; the approach by Bi and Hieronymi (2024), which distinguishes ten classes,





is ideal for this. Alternatively, other OWT frameworks can serve as a basis (e.g. Vantrepotte et al., 2012; Moore et al., 2014; Mélin and Vantrepotte, 2015; Jackson et al., 2017; Spyrakos et al., 2018; Bi et al., 2021), but these are often more focussed on

inland waters or oceans, and in some cases up to 17 classes are differentiated, which makes it very difficult to find class-representative in-situ data. Ideally, one can utilise matchups in which all parameters were measured simultaneously, e.g. as on the dedicated research cruise in the Baltic Sea in July 2023 (Hieronymi et al., 2023b) - but this also only covers one water class (OWT 4a). We aim to utilise new data handling technologies, such as the AquaINFRA Data Discovery and Access System (https://aquainfra.eu/), which allows researchers to seamlessly find and retrieve datasets from heterogeneous sources, e.g. from

stationary or moving measurement platforms, from long time series or targeted experiments.

In perspective, this dataset and the underlying algorithms are an opportunity for the exploitation of remote sensing in order to answer limnological-oceanographic questions and expand monitoring capabilities. This could, for example, be a contribution to observation-based carbon budgeting (e.g. Friedlingstein et al., 2023; Brewin et al., 2023), whereby coastal DOC and POC are already provided and there is a good starting point for PIC and primary production. As has been shown, the optical water

types are particularly useful as indicators for phytoplankton groups, and there is potential to better assess the phytoplankton function in the ecosystem, contextualising long-term observations, or to warn of harmful algal bloom (e.g. Bracher et al., 2017; Kordubel et al., 2024; Devreker et al., 2025).

## 8.  Data availability, visualisation, and context

### 8.1. Data availability

The dataset consists of 122 day-aggregated single NetCDF files totalling 365.16 GiB. It can be freely obtained from the World Data Center for Climate at https://doi.org/10.26050/WDCC/AquaINFRA_Sentinel3 (Hieronymi et al., 2024). Identical Sentinel-3 OLCI-based datasets with the same parameters have been produced for testing purposes for other regions of the world and will be made available soon at the same database: 1) the Mackenzie River region with the Beaufort Sea and Arctic Ocean, 2) the Black Sea with the Sea of Marmara and part of the Aegean Sea, and 3) the central North Atlantic.

### 8.2. Data visualisation

The NetCDF data can be opened and visualised in the Sentinel Application Toolbox SNAP or in QGIS, for example. In addition, there is a visualisation of the content using cloud-optimised GeoTIFFs, which was provided by the EU AquaINFRA project partner *Finnish Geospatial Research Institute* at the *National Land Survey of Finland*. This can be used to quickly check the actual availability of cloud-free water quality parameters of the Sentinel-3 OLCI ONNS dataset in two projected

coordinate systems, namely

Pseudo Mercator https://vm4072.kaj.pouta.csc.fi/ddas/oapic/collections/sentinel-3-OLCI-ONNS3857  and

LAEA https://vm4072.kaj.pouta.csc.fi/ddas/oapic/collections/sentinel-3-OLCI-ONNS3035.





### 8.3. Corresponding Sentinel-3 OLCI data and additional resources

Several data resources are available where partly the same Sentinel-3 OLCI input data are interpreted (processed) in a different
way or where data from other satellite missions are visualised. The Copernicus Data Space Ecosystem Browser serves as a
central hub for accessing, exploring and utilising Earth observation data such as from Sentinel-3 OLCI, where the standard
Level-2 water products as well as many synergetic products can be viewed (https://dataspace.copernicus.eu/browser/). The
Copernicus Marine Service provides physical and biogeochemical reference information on the state of the oceans, partly in a
regionally optimised manner for Sentinel-3 OLCI with specific products for the North Sea, Baltic Sea, and coastal waters –
but also in the context of other European seas (https://marine.copernicus.eu/). The Copernicus Land Monitoring Service
provides information on land cover and land use, the water quality of lakes, and the water level of lakes and rivers in the
hydrographic network worldwide and in Europe (https://land.copernicus.eu/en). The US-American National Oceanic and
Atmospheric Administration (NOAA, NESDIS, Center for Satellite Applications and Research) hosts a valuable ocean colour
viewer that puts Sentinel-3 OLCI data in context with other global ocean colour missions, most notably from the Visible
Infrared Imaging Radiometer Suite (VIIRS) on two satellites (https://www.star.nesdis.noaa.gov/socd/mecb/color/index.php).
The Finnish Environment Institute (Syke) operates the Tarkka service with a map viewer for satellite images of the Baltic Sea
region and parts of the North Sea, which combines Sentinel-3 OLCI images with even higher spatial resolution Sentinel-2 MSI
and Landsat images (https://tarkka.syke.fi). Sentinel-3 OLCI data are also integrated into global long-term time series, e.g. as
part of the ESA Ocean Colour - Climate Change Initiative with observations since 1997 (https://www.oceancolour.org/), where
spatiotemporal changes of water constituents in the oceans can be monitored (e.g. Oostende et al., 2023).

### 9. Conclusions

We provided a satellite image-based dataset that includes water quality properties in lakes, rivers, coasts, as well as the entire
North Sea and Baltic Sea. We applied a novel data processing scheme with the aim of seamless and consistent data quality in
the transition between the optically diverse waters. The analysis of the optical water types shows clear differences between the
water bodies, which, in addition to seasonal phytoplankton phenology, are governed by CDOM absorption or hydrodynamic
sediment suspension. CDOM is mainly discharged into the sea via the freshwater of inland waters and diluted in the sea; the
salinity of water is therefore a good indicator for optical nuancing and shows, for example, a clear difference between the
ocean/North Sea and the Baltic Sea. Optical water type classification helps to specify water constituents more precisely and
serves as a direct hint for some phytoplankton groups. With this dataset, we present for the first time our approach for IOP-
based estimation of particulate and dissolved organic matter. Currently, we only provide a quantitative grading of the water
quality parameters, which vary over several orders of magnitude – a detailed and OWT-specific validation of the 73 individual
parameters to must follow. Overall, however, the dataset offers many starting points for deriving further parameters and user-
oriented information from the remote sensing data in the future.






**Author contributions**

MH designed the research, compiled and analysed the dataset, and wrote the manuscript. BD, SB, and RR contributed to the fundamental research, compiled and processed the dataset, and reviewed the manuscript.

**Competing interests**

The authors declare that they have no conflict of interest.

**Acknowledgements**

This work is a contribution to the EU project AquaINFRA, which has received funding from the European Commission's Horizon Europe Research and Innovation programme under grant agreement No. 101094434. The analytical work is a contribution to the PHY2FLEX project, which was funded by the European Space Agency (ESA contract ref.
4000147260/25/NL/SD). Additional support was provided by the German Helmholtz Association with the research program "Earth and Environment" (PoF-IV) and the Hereon-I$^2$B project PhytoDive. The work is based on free and open satellite data from the European Union's Copernicus Programme provided by ESA and EUMETSAT. which is highly appreciated. We would like to thank Therese Harvey (NIVA, Danmark), Eileen Hertwig (WDCC, Germany), and Björn Saß (HEREON, Germany) for user feedback, FAIR data preparation, and valuable discussion. Moreover, we would like to thank or Finish
partners Lassi Lehto, Jaakko Kähkönen, Hanna Lahtinen, and Pekka Latvala (FGI, Finland) for data visualisation and support. Thanks also to Eike Schütt (Kiel University, Germany) for his contributions to the compilation of the climatological data.

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
