# Peer review of "Optical complexity of North Sea, Baltic Sea, and adjacent coastal and inland waters derived from satellite data"

_Earth System Science Data, 2025_

## Author Comment (AC2)

**RC1: 'Comment on essd-2025-443', Anonymous Referee #1, 06 Nov 2025**

We would like to thank you very much for reviewing our work and for your detailed comments. We will endeavour to incorporate your comments into the manuscript, primarily with a view to clarifying the discussion.

But first, we would like to comment on the main comment on the issue of validation, which all three reviewers found lacking. Working on an end-to-end processor for satellite imagery is very complex and time-consuming, it involves an atmosphere and a water component with strong interdependencies. For this reason, it has become apparent that the entire system should be considered for optimisation, also to develop appropriate flags. There are five fundamentally different data groups that need to be validated: meteorological (wind, solar irradiation), oceanographic (water temperature and salinity), apparent optical properties ($R_{rs}$, $K_d$, $K_u$, $FU$), inherent optical properties, and concentrations of water constituents - each sub-parameter has its own measurement method and error acceptance (e.g. IOCCG report on uncertainties, 2019). Furthermore, measurement methods encounter (undefined) limitations when used in different optical water types, e.g. $R_{rs}$ measurements below or above water in clear or very turbid water. The call for product validation is justified, but it cannot be realised in a simple and compact manner in this paper. With this work, we aim to explain the fundamental assumptions, methods, and processing steps using a test dataset (and a scientific question about optical complexity of our region of interest) to also enable user feedback and, thus, create a citable reference to the community.

In fact, some of the products have already been compared with in situ data in the mentioned regional studies, with A4O-ONNS performing unsatisfactorily in some cases. The focus of the analyses in this article refers to remote-sensing reflectance in 14 out of 16 spectral bands, which serve as the basis for OWT analysis. In this regard, we would like to refer to the study by Hieronymi et al. (2023), in which $R_{rs}$ from A4O was compared with in-situ data, particularly in comparison with the established atmospheric correction methods IPF, C2RCC, POLYMER, and Acolite. The comparison, e.g. here in Fig. 4 from their publication, also underlines the importance of flags for valid pixel expression. For example, in the standard AC of OLCI (IPF), large areas are conservatively flagged out, e.g. in sun glint (albeit the results may actually look good). Other methods such as A4O have no restrictions here so far; this alone results in significantly more possible matchups. In the processing currently under discussion, results from high sun glint areas are also incorporated into the Level-3 daily mean values. In preliminary tests, we have not found excessive deviations, but the test dataset serves to precisely analyse products and to define the valid pixel expression. For monitoring of the land-sea transition, the fundamental provision of classifiable $R_{rs}$ spectra for all defined water types is crucial, and this is where A4O offers clear advantages. To underpin this, we would like to draw attention to Fig. S10 from the Supporting Information for Bi and Hieronymi (2024) in comparison with Fig. 12 of our article.

In fact, we are currently working on another publication that compiles all available $R_{rs}$ measurements from the same region and time to perform validation of this dataset. That said, the data is very diverse from AERONET-OC or WATERHYPERNET stations, ship validation campaigns, autonomous ferry ship measurements, and measurements taken above or below water. Much of the available data does not meet *Fiducial Reference Measurements* quality requirements, and comparative values are not available for all water types. Describing these methods and their differences is complex and lengthy, and beyond the scope of this paper here. Please also note that measurement uncertainties for $R_{rs}$ close to zero are very high, i.e. generally

in the NIR, but also in the blue at higher CDOM concentrations, this limits meaningful validation for the bands. We would therefore suggest that the references to previous validation studies be highlighted more clearly in the manuscript.

[Figure]

Figure 4 from Hieronymi et al. (2023) showing a comparison of satellite-derived $R_{rs}$ with AERONET-OC in situ data for OLCI bands at 412, 490, 560, 665 and 865 nm for IPF (A–E), C2RCC (F–J), A4O (K–O), POLYMER (P–T), and ACOLITE-DSF (U-Y). The colors represent different stations. The contours indicate the density distribution. The different numbers of matchups ($N$) are mainly due to method-specific flagging and the spatial homogeneity criterion per band.

[Figure]

Figure S10 from the Supporting Information for Bi and Hieronymi (2024). It shows classifiability of $R_{rs}$ from five atmospheric correction methods (including A4O, C1-C3) based on ten diverse OLCI test scenes.

The only water parameter really discussed in the article is particulate organic carbon concentration. We therefore suggest using this parameter to describe the complexity of validation and adding the following section:

"There is only very little measured and uncertainty-characterized validation data available for the same region and time. From a validation campaign with *RV Alkor* (AL597) in July 2023 in the Baltic Sea *POC* measurements are available with concentrations varying between 0.25 to 1 g m$^{-3}$ (*N* = 48). The related determination errors were generally low (1-5 %), but in few cases up to 20 % (Hieronymi et al., 2023b; Novak & Röttgers, 2026). Based on both measured and satellitederived reflectance, the waters during all these measurements were assigned to just one optical water type, namely OWT 4a. Nevertheless, there are considerable small-scale spatial variations, often caused by the occurrence of large cyanobacteria colonies. Standard matchup criteria require a narrow time window and spatial homogeneity, but cloud cover also limits the amount of available comparison data. To estimate retrieval performance, we use all measurement points and corresponding satellite observations for the entire month of July 2023. Figure 15 shows a comparison of *POC* values from satellite observations with median of values from 3x3 pixels using a homogeneity criterion and cloud-free conditions within 5x5 pixels. The error bars for satellite-derived *POC* represent the minima and maxima of the mean values for the month and indicate that the values can vary by an order of magnitude, which is often related to unrecognized cloud artefacts (typically resulting in higher *POC* concentrations). This shows that rigorous system-wide flagging needs to be improved. However, Pearson's correlation coefficient shows a strong positive relationship (*r* = 0.76). The absolute root-mean-square error (*RMSE*) is 0.25, the mean absolute error (*MAE*) is 0.23, and the median absolute relative difference (like in Smith et al., 2018) (*MARD*) is 45 %. Linear regression yields a slope of 0.6 and an intercept close to zero; *POC* is therefore rather underestimated with considerable variability. However, we argue that explicit uncertainty characterization is also the dimension of high-quality earth observation data. This rough comparison (for one water type) is similar to the other OC products in the dataset and shows that there is still substantial need for improvement of the end-to-end processor. However, it also shows the opportunities for OWT-specific adaptation of IOP-concentration relationships and application of system vicarious calibration."

[Figure]

New Fig. 15: Comparison of particulate organic carbon measured in-situ from water in the Baltic Sea against satellite estimates from the same month (July 2023). The horizontal error bars of the in situ *POC* concentrations represent the standard errors from the determination. Vertical lines show the variation in median values in 3x3 pixels of the satellite images over the entire month. The 1:1 line is shown as a dotted line.

**RC1**: The manuscript documents a daily aggregated, Level-3 dataset produced by merging all available Sentinel-3 OLCI observations from both S3A and S3B overpasses for each day across the North Sea and Baltic Sea during June to September 2023. The processing chain uses the

A4O atmospheric correction followed by the ONNS neural network water processor. The data product includes remote-sensing reflectance at 16 OLCI bands, a suite of inherent optical properties, concentrations such as chlorophyll, TSM, DOC and POC derived from IOPs, the Forel-Ule color index, optical water type classifications, and several quality and context flags including cloud masks, adjacency risk, glint risk, bright pixel flags, and a whitecap fraction parameterization. The paper states that this is a prototype release, that no full validation of the many variables is provided here, and that the code will only be released in the medium term. The archived dataset is now at WDCC with a DOI, CC-BY 4.0 license, and a variables document.

The dataset targets a well known gap in ocean color for optically complex waters at the land sea interface where standard processing is often limited. Using both S3A and S3B improves daily spatial coverage, and a single pipeline across inland and coastal waters is attractive for monitoring and for synoptic biogeochemical analyses. Coupling an OWT framework to both atmospheric correction and water property retrieval is methodologically coherent, and providing OWT outputs alongside geophysical variables is useful for quality screening and for science use. As an ESSD submission, the core value is the accessible, gridded Level-3 product with metadata, DOIs, and a usage context. These positive aspects are clear in the paper.

**Response**: Thank you for recognising the value of our work and highlighting its potential.

The manuscript explicitly states that publication of the code is only planned in the medium term and does not include a Code availability section. ESSD allows data-only descriptions, yet it strongly encourages deposition of software and algorithms in FAIR repositories and requires a Code availability section when code is part of the work. For complex EO processing pipelines that strongly condition the resulting data, ESSD policy emphasizes transparency and reproducibility as core principles. In its current form, the work falls short of these expectations because independent users cannot reproduce the dataset or verify implementation details of A4O or the specific ONNS configuration used. At minimum, a versioned, citable container image or repository with the exact A4O and ONNS code paths, trained weights, and runtime environment is needed, together with a Code availability section that points to those DOIs.

**Response**: Thank you for your comment – we agree in principle. There are several aspects to consider here. Firstly, it is primarily about a dataset that we have created with great effort and many days of runtime; that is a value in itself. We have cited several sources with alternative processing methods (section 8.3), some of which are based on the same original data; in this respect, comparisons are possible and transparency is guaranteed. Please also note that many of the Level-2 or Level-3 methods mentioned are not freely available, including the standard processing by EUMETSAT. We are actively working on improving the technology readiness level of the end-to-end processor. This paper serves to document the basic concept, albeit in an intermediate stage, where the AC, OWT, water components and flagging have not yet been adapted. The development of this complex system is dynamic, and substantial changes are foreseeable. We are currently unable to publish the code for the processing chain as it stands, but we will prioritise after your feedback (depending on funding).

Please also note that this article primarily deals with the scientific question of the optical complexity of the North Sea-Baltic Sea region. As described in Hieronymi et al. (2023), available atmospheric corrections are insufficient to represent the complexity across different OWT frameworks, because they do not provide $R_{rs}$ of some defined classes, i.e. they do not function well for all waters. Based on the findings of this study, a new OWT framework (Bi and Hieronymi, 2024) was created, its code is freely available (https://github.com/bishun945/pyOWT). As illustrated in the figure above from their publication, A4O generates $R_{rs}$ spectra with the widest

optical variance and the best classifiability. In principle, however, we could have clarified the question of optical complexity of the region using any other AC method; the uncertainties were explained in the article, but the results should be fundamentally comparable.

**Changes**: We include a reference to the OWT code, which can be used to reproduce the results of this study or can be applied to other atmospheric correction methods or satellite data.

The retrieval chain uses neural networks and an AC method. Small choices in training data, preprocessing, and band handling materially change IOPs and derived concentrations. Without the code or at least a fully specified ATBD with the exact trained model artifacts, an independent group cannot regenerate the L3 product from S3 Level-1 data. That limits reuse and undermines the central ESSD promise of transparent, reusable data products.

**Response**: As already noted, providing codes and training data is a complex task that is not feasible for this publication. The data used to train the neural networks – and this is only one aspect of the overall system – consists mainly of results from extensive solar radiative transport simulations in the atmosphere and in water. Processing the large amounts of data would be a huge task, but publication will be considered in the future – now, it cannot be made available. We have also written that there will be further adjustments regarding OWT-specific water NNs. Biogeo-optical modelling has developed significantly, especially with regard to assumptions for particulate scattering (Bi et al., 2023).

We would also like to refer back to our discussion on the dataset. It offers a wide range of parameters and links for further application of all kinds of "algorithms", such as deriving chlorophyll concentration from reflectance or determining Secchi depth from $K_d$. One could therefore apply any (!) *Chl* algorithm and compare it with our *Chl* estimate. In principle, one could even apply a different atmospheric correction and subsequent water algorithms to our dataset, as it contains top-of-atmosphere reflectance too. We have explicitly placed great emphasis on reusability, transparency, and connectivity of the dataset.

The ONNS basis is documented in Frontiers in Marine Science and is citable, which is a strength. However, the present chain departs from the 2017 ONNS in key ways. The paper indicates that concentrations now come from ONNS-derived IOPs rather than directly from class-specific networks, and that the OWT scheme used here is the newer Bi and Hieronymi framework. Those choices are reasonable, yet they change the forward model and error propagation, so they must be documented with enough specificity to be reproducible. The A4O method has been compared against other ACs, but a full methodological description plus code or trained models are still not publicly archived.

**Response**: We acknowledge that publishing a dataset without the full, open-source processing chain presents challenges for direct reproducibility. However, we argue that the primary contribution of this work lies in the unique coverage and quality of the generated products, particularly in optically complex waters where standard processors often fail. This dataset provides a critical benchmark that allows the community to evaluate the results, even while the full processing tool remains under development.

Regarding the selection of output variables, our approach mirrors established operational practices. For instance, standard Sentinel-3 OLCI Level-2 products provide both CHL_OC4ME (semi-analytical) and CHL_NN (neural network) estimates. This is not an inconsistency, but a recognition that different algorithms rely on different assumptions, and their performance varies depending on the optical water type. By providing our specific ONNS-derived IOPs and concentrations alongside standard products, we ensure transparency in our method's

performance. This approach allows users to evaluate the "fitness for purpose" of different algorithms for their specific regions of interest and to report which product yields the most accurate representation. Therefore, this dataset serves as a necessary intermediate step to enable such comparative studies and user feedback, which are essential for the future harmonization of the end-to-end processor.

The manuscript is explicit that it does not perform a full validation of the many variables. For an ESSD data description that is acceptable only if adequate demonstration of fitness for purpose is provided and if uncertainties and quality information are delivered in a way that users can apply. Here, the validation evidence is mostly qualitative, which is a weakness. The paper even notes a possibly erroneous blueward tendency of A4O in some conditions and that reflectance magnitude is often underestimated, which is significant because Rrs is the driver for all IOP and concentration products. Users need at least some quantitative, OWT-stratified matchup statistics versus in situ Rrs and against IOP and concentration measurements, with uncertainty budgets that follow accepted EO data record practice. A concise validation plan can be staged, but the first ESSD version requires some validation.

**Response**: Please note our comment at the beginning and the difficulties involved in carrying out comprehensive validation. However, beyond standardised product validation, which is indeed lacking, we would like to draw attention to the detailed OWT analysis in the ESSD manuscript. This demonstrates precisely the fitness for purpose in a way that classic validation does not. The above Fig. S10 from Bi and Hieronymi (2024) shows the ability of five atmospheric correction methods for OLCI to produce certain spectral shapes and magnitudes of $R_{rs}$. Compared to standard AC (IPF collection 3), A4O never outputs negative reflectance, is not initially flagged as invalid in high sun glint, and induces less spatial noise, and thus generates "valid" results for an image area twice as large – with generally better classifiability, i.e. accepted $R_{rs}$ spectra (Hieronymi et al., 2023). IPF has for example problems in cases with high algae biomass (OWT 5a&b) or high *CDOM* concentrations (OWT 7), which would be particularly important for the Baltic Sea and inland waters. Figure 12 from our manuscript shows that $R_{rs}$ from A4O can be well-classified in 99.68% of the cases in this region. In the absence of dense in-situ networks, such spectral consistency and classifiability serve as critical proxy metrics for data quality. These are arguments that prove that A4O has a better fitness for purpose for this optically complex region than the standard atmospheric correction, which is considered validated. The uncertainties of the classification were outlined and reflected in your remarks; improving performance is a subject for ongoing research. Once again, we would like to point out that with this published dataset and its description in ESSD, we can now carry out careful OWT-specific validation of all 73 products (where available).

**Changes**: We propose further elaborating on the aspect of fitness for purpose and formulating a clearer validation plan.

The variables list in the paper and on WDCC is helpful, but several names and units would benefit from alignment with existing community standards. For NetCDF, CF conventions recommend using standard_name attributes where possible and consistent units and descriptive long_name fields. For ocean color, ESA CCI and NASA ocean color products provide a de facto vocabulary, for example RRS for remote-sensing reflectance, CHLOR_A for chlorophyll, K_490 for diffuse attenuation at 490 nm, APH for phytoplankton absorption, ADG for CDOM-plus-detritus absorption, and BBP for particulate backscattering. The present ONNS variable names such as ONNS_a_g_440, ONNS_b_p_510, and the use of the term Gelbstoff for

CDOM are understandable in context but may confuse users who expect CF-style names and common ocean color acronyms.

**Response**: The naming of variables is a serious problem, where we spend much time trying to solve it. Within the mentioned sources in Section 8.3, the following products for chlorophyll concentration based on OLCI-estimates are used: CHL_OC4ME, CHL_NN, Chl-a, Chlorophyll-a, chlor-a, chla_mean, chl, CHL, etc. Even for such common parameters as chlorophyll concentration, each source uses its own terminology. A genuine standard, e.g. at IOCCG level, would be desirable. Also, the use of "K_490 for diffuse attenuation at 490 nm" is too unspecific, at least Kd_490 with *d* for *downwelling* is useful. We feel that many "CF style names" are not precise enough defined in our field and cross-cutting limnology and oceanography. Many definitions are misleading https://cfconventions.org/Data/cf-standard-names/current/build/cf-standard-name-table.html (e.g. search for chlorophyll) - none of the standard names refer to fresh water for example. Our naming of variables is guided by the usage of the software Hydrolight and Mobley (1995). The names may not be ideal, but they follow a structure that we consider reasonable, e.g. naming the concentration and the underlying method at the same time, as in CHL_OC4ME. With IOP_Chl, we emphasise that the concentration is derived from the IOPs and not directly from a neural network, for example.

**Changes**: We include a discussion element for future initiatives.

On reflectance terminology, the manuscript lists A4O_Rrs_n as normalized remote-sensing reflectance. In ocean color there is potential confusion between fully normalized water-leaving radiance nLw, remote-sensing reflectance Rrs, and various normalization schemes. The paper should define exactly what normalization means in A4O, how it differs from standard Rrs, and why the units remain sr-1. That definition should also be embedded in the NetCDF metadata so that users do not misinterpret the quantity.

**Response**: Here you address aspects that also illustrate why the validation of the diverse product groups cannot be dealt with on a single page, but rather deserves dedicated, complex studies. The atmospheric correction A4O approximates a remote-sensing reflectance (bottom-of-atmosphere) from directional radiance and transformed reflectance at the top-of-atmosphere under the following characteristics: 1) The sun's position and viewing angle are assumed to be exactly perpendicular ($\theta_S$ = 0°, $\theta_V$ = 0°), 2) wind influences are set at 5 m/s, which is a standard assumption for water algorithms, and 3) $R_{rs}$ is free from the effects of whitecaps and air bubbles in the water. These are harmonised angles and environmental conditions to enable global comparability and maximise the exploitation of satellite data, including in sunglint conditions. The disadvantages are that measured reflectance must be modified quite significantly to obtain fully normalised reflectance and to ensure comparable conditions, and that these modifications depend on the optical water types. The uncertainties of the approach and assumptions are reasons why sensitivity tests based on such a dataset are necessary.

**Changes**: The precise definition of the delivered $R_{rs}$ and the boundary conditions are integrated.

The provision of flags is welcome, including cloud masks, cloud risk near edges, adjacency, glint risk, bright pixels, and a special flag for very high biomass or floating algae. The inclusion of a whitecap fraction parameter (A4O_A_wc) is scientifically useful because whitecaps increase broadband water-leaving signal and can bias retrievals if not handled. The whitecap parameterization is cited to satellite-based work, which is appropriate. What is missing is a clear, file-embedded description of how users should combine these flags for robust quality screening and what the recommended filters are for computing spatial or temporal aggregates.

Given that the paper acknowledges artifacts near clouds and adjacency and a blueward bias in some regions, the dataset should come with a documented, conservative quality mask and a short tutorial for users.

**Response**: Good point. The dataset is intended to be used to provide better recommendations for flags. Some masks can be regarded as independent parameters, e.g. cloud cover and floating algae, which also detects Sargassum, for example. Masks should always be seen in their spatial context; in the Baltic Sea, floating algae also marks intense Cyanobacteria blooms with a high Rrs-NIR signal, where the basic assumptions can be problematic if one cannot see into the water but estimates the concentration in volume. With regard to the whitecap fraction, it is a feature of A4O that these effects are removed so that subsequent water-retrieval is not biased. In this respect, the specification serves not only to provide information on air-sea fluxes, but also to ensure transparency that potential influences have been removed.

**Changes**: Recommendations for using the flags are included.

The dataset is built from both Sentinel-3A and Sentinel-3B OLCI sensors merged to daily Level-3. That is effectively a dual-sensor product within a single mission. The title reads as derived from satellite data, which could be interpreted as multisensor across missions. The abstract clarifies that the source is Sentinel-3 OLCI, and the methods section explicitly states S3A and S3B. To avoid misunderstanding, I recommend reflecting the instrument in the title or at least stating prominently on first mention that this is an OLCI-only product that merges S3A and S3B.

**Response**: We focus on the scientific question of the optical complexity of the region in order to provide a benchmark for the reliability and comparability of the algorithms used, e.g. for the Copernicus Marine or Land Services. This could also be estimated from other satellite data (MODIS, VIIRS, PACE, multi-sensor merged, or from diverse Copernicus Services) or widely distributed in situ reflectance measurements. Similar spatial patterns are also shown in Mélin and Vantrepotte (2025) based on SeaWiFS, for example. Sentinel-3 OLCI is a well-suited platform for this, but we would not want to over-emphasise this in the title.

**Summary**

ESSD requires a Data availability section and encourages authors to archive software and provide a Code availability section. The paper satisfies data availability through the WDCC DOI. It does not yet satisfy the spirit of ESSD reproducibility for algorithmic data products, because the code is not accessible and the AC is not documented at the ATBD level. ESSD explicitly invites authors to deposit code and even supports literate programming submissions to maximize transparency. This manuscript should follow that guidance for acceptance. The dataset fills a scientific gap but the present paper is not ready for acceptance because reproducibility and quantitative validation are not yet sufficient, and because naming, terminology, and user guidance need revision for broad reuse. If the authors release the processing code, add validation and/or uncertainty descriptions, align variable metadata with CF and common ocean color practice, clarify scope, and document flagging rules, I would recommend acceptance after those changes.

**Response**: In this paper, we have interwoven two aspects: 1) description of a satellite data processing chain with A4O-ONNS and 2) analysis of OWT in the region based on an A4O-ONNS dataset. The larger part is about OWT analysis, and with the suggestions of the other reviewers, it is growing even more. The OWT code is freely available. The OWT analysis can also be based on completely different data, as long as the methods used are fit for purpose. This has been demonstrated with the A4O atmospheric correction and OLCI data. Reproducing the underlying

data set is indeed not yet possible for outsiders, but it would also involve a massive effort and, in our opinion, would not be necessary to clarify the scientific question addressed in the paper. We have openly communicated the further development steps of the overall algorithm in the document. However, we have also shown in our responses here how complex and complicated your demands would be. These questions cannot be answered in a few extra pages and are therefore out of scope for this work. We will document and validate the individual aspects of the processor separately, e.g. atmospheric correction and the corresponding $R_{rs}$ as output. This requires a thorough understanding of optical water types, also to understand measurement errors. This also offers the possibility of targeted application of System Vicarious Calibration per OWT. Thus, we hope that our changes and exemplary comparisons with (relatively) few uncertainty-characterised measurements (of $POC$) will lead to acceptance of our work.

**References:**

Bi, S., and Hieronymi, M.: Holistic optical water type classification for ocean, coastal, and inland waters, Limnol. Oceanogr., 69 (7), 1547-1561, https://doi.org/10.1002/lno.12606, 2024.

Hieronymi, M., Bi, S., Müller, D., Schütt, E. M., Behr, D., Brockmann, C., Lebreton, C., Steinmetz, F., Stelzer, K., and Vanhellemont, Q.: Ocean color atmospheric correction methods in view of usability for different optical water types, Front. Mar. Sci., 10, 1129876, https://doi.org/10.3389/fmars.2023.1129876, 2023a.

Hieronymi, M., Röttgers, R., Alikas, K., Kratzer, S., Ansko, I., Behr, D., Bi, S., Burmester, H., Heymann, K., Novak, M., Pramlall, S., Rahn, I.-A., Roux, P., Thölen, C., and Voynova, Y. G.: Biogeo-optical and-chemical characterization of cyanobacterial blooms in the Baltic Sea for ocean colour satellite remote sensing, Alkor Cruise Report No. AL597, 24 pp., https://doi.org/10.3289/CR_AL597, 2023b.

IOCCG: Uncertainties in Ocean Colour Remote Sensing, Mélin F. (ed.), IOCCG Report Series, No. 18, International Ocean Colour Coordinating Group, Dartmouth, Canada. http://dx.doi.org/10.25607/OBP-696, 2019.

Mélin, F., and Vantrepotte, V.: How optically diverse is the coastal ocean?, Remote Sens. Environ., 160, 235-251, https://doi.org/10.1016/j.rse.2015.01.023, 2015.

Novak, M. G. and Röttgers, R.: Spectral variability in particulate light backscattering in the sea: Resolving fine-scale optical features and their biogeochemical significance, Opt. Express, 2026 Accepted/In Production.

Smith, M. E., Robertson Lain, L., and Bernard, S.: An optimized chlorophyll a switching algorithm for MERIS and OLCI in phytoplankton-dominated waters, Remote Sens. Environ., 215, 217-227, https://doi.org/10.1016/j.rse.2018.06.002, 2018.

---

## Author Comment (AC3)

**RC2: 'Comment on essd-2025-443', Timothy Moore, 29 Dec 2025**

Thank you very much for carefully reviewing our paper and for your helpful comments on how to improve it. Your contributions are very much appreciated. We will address all points below (our responses are highlighted in blue).

**General comments**

Overall: The present manuscript describes an end-to-end processing chain for generating regional ocean color products from a processing scheme that incorporates flexible choices for atmospheric correction and bio-optical algorithms through a classification scheme developed in a previous manuscript. This comprehensive approach is a realization of how optical water type classification schemes can be fully utilized in a processing chain, as both integrating intermediate products for blending, and as stand-alone products in of themselves. The developed scheme covers a wide variety of waters, from inland systems to coastal and open oceans of the Baltic/North Atlantic region. The developed scheme addresses problems of choosing algorithms across such diversity of systems.

**Response**: We appreciate your positive assessment of the benefits of this new technology.

The presentation of the overall manuscript is reasonable. However, notable omissions hinder important aspects of the manuscript. First, the OWT scheme, which is the central 'glue' to the processing chain, is not visualized and referenced in a previous manuscript. The OWTs are never shown or clearly explained as to what they represent. A new section should be devoted to this. What are the centroids and distributions of the 3 variables - this could be shown in 3-d space. What do the Rrs shapes look like, what are the spectral/optical characteristics, why are there 'a' and 'b' subdivisions - are these variations? I think it is crucial for a table and/or figure showing the OWTs, as these are referenced throughout the paper.

**Response**: Thank you, it's a good recommendation for introducing the OWT framework even better.

**Changes**: We have written a separate chapter on the OWT system (new 2.4) and adapted previous information accordingly. We have also added an illustration of the mean spectra, the co-variances of the optical parameters and a brief description (new Fig. 4).

[Figure]

Figure 4: Spectral characteristics and definitions of the Optical Water Types (OWTs) from Bi and Hieronymi (2024). (a) Mean remote-sensing reflectance ($R_{rs}$) spectra for each type. (b-c) Distribution of OWT clusters across the three optical classification variables: Apparent Visible Wavelength ($AVW$), Spectral Magnitude ($A_{BC}$), and Normalized Difference Index ($NDI$). (d) Descriptive summary of the ten water types.

Secondly, an overarching figure showing the scheme would also be very helpful. The 'algorithm' is really a fully developed scheme that incorporates A/C with classification and production of in-water properties as products. Using the term 'algorithm' to describe the full processing chain is somewhat mis-leading, as many readers may associate 'algorithm' with a formula for generating a specific bio-optical product from Rrs.

**Response**: A good suggestion to emphasise the complexity of the end-to-end processor more strongly.

**Changes**: We have expanded Fig. 2 and added the proposed workflow of the overall processing chain.

[Figure]

Lastly, the final optical products produced are not validated with any field data, but are qualitatively assessed. Operationally, extra processing effort (=time and computing power) is required for such a processing chain, which is justifiable if this leads to product improvement. This lack of verification without comparisons with 'validation' data leaves this question dangling. While this has been demonstrated in other earlier OWT-based studies, it is left unanswered here.

**Response**: Please refer to our detailed comments on validation in response to Reviewer #1. In summary: The validation of the 73 individual parameters in the dataset, including $R_{rs}$, IOPs, and concentrations of water constituents, is very complex and involves specific error assessments. In addition, we call for an OWT-specific evaluation of "algorithm" performance, as measurement methods are also not applicable to all water types. We would like to refer more prominently to earlier work and comparisons, especially for $R_{rs}$, which is the basis for OWT analysis. We are aware of the pressing issue of validation and want to address it more thoroughly, but not comprehensively in this article. We propose to address the validation issue using the one water parameter described, namely *POC*, where there is very little measurement data that matches the satellite data set. This comparison is only meaningful for one water class, not for the other nine defined OWTs. Our main focus is on demonstrating the fitness for purpose of the A4O atmospheric correction for this region, which includes all defined OWTs.

The goal of this development is to create a processing system that works quickly, efficiently, and provides reliable products, making it suitable for operational use. By using OWT-specific water algorithms, we want to improve water product quality (because we can reduce ambiguities and adjust the scope), but this requires additional computational effort. We are currently working on a more efficient solution and system integration; validation of the end-products will follow.

**Changes**: We complement the quantitative evaluation of the POC results with a comparison with in situ data from the Baltic Sea and July 2023. Moreover, we add discussion on a validation plan.

**Specific Comments:**

L29: PACE now expands wavelength range into the UV

**Changes**: UV included.

L44: What is meant by the 'view'? Is this a geometric view from satellite, or a philosophical view of ocean color?

**Changes**: We reformulated the sentence to "Earth observation of large areas with different types of waters also poses significant challenges for analysing satellite images."

L98-104: Not familiar with the geographic landmarks mentioned and they are not indicated on the map.

**Changes**: The overview map (Fig. 1) has been modified accordingly and the waters mentioned in the text have been labelled.

[Figure]

L124: so, 4 months of image data used in total?

**Changes**: Yes, "four-month period" was included.

L134: What is meant by the 'algorithm'? It sounds like at least part inclusion of an atmospheric correction? Should this be the 'processing scheme' or 'algorithm scheme'? Also, The 'used' algorithm could be phrased better...the 'developed' algorithm or 'proposed algorithm'? A schematic figure showing the processing chain and/or evolution of the 'algorithm' would be a useful (critical!) figure for this section.

**Response**: Thanks for the hint to unclear wording. We will emphazise more the "processing scheme" or "end-to-end processor" in the manuscript.

**Changes**: We changed the wording and included a schematic figure illustrating the workflow.

L153: the 'used processing chain' is a bit awkward phrasing...alternative suggestions: the 'developed processing chain'...the 'implemented processing chain'...the 'presented'. This section is a bit confusing because now it appears that daily averaging is part of the 'algorithm', along with in-water production generation. I really think this needs to be clarified what components are involved in the 'algorithm' - A/C correction, water typing, averaging, product generation (a schematic would help here).

**Changes**: Thanks for the suggestions, we rephrased it accordingly.

L155-160: Unclear how many pixels were used for the developing the OWT scheme.

**Response**: We apply the OWT scheme developed by Bi and Hieronymi (2024) based on simulated $R_{rs}$ data, which are based on state-of-the-art biogeo-optical modelling by Bi et al. (2023). Thus, the OWT scheme is independent of any actual satellite data but can be flexibly applied to atmospherically corrected scenes with many different band configurations, e.g. for

Sentinel-2 MSI, VIIRS, PACE etc.. A fundamental description of the OWT scheme has been included.

L165-238: Elements described here could be better visualized if included in an overarching schematic.

**Changes**: We included a schematic figure illustrating the workflow.

L252: 'already favoured'->'promoted' or 'advanced'?

**Changes**: We chose "promoted".

L270: Comment: the OWT scheme is based on satellite data (it was unclear up to this point).

**Response**: No, actually not. Both independent OWT schemes, Hieronymi et al. (2017) and Bi and Hieronymi (2024), are based on radiative transfer simulations and resulting $R_{rs}$. The mentioned publication by Hieronymi et al. (2023) compared four different OWT schemes applied to atmospherically corrected OLCI scenes.

L275: This is an important section and its a bit unclear how or what the premise of the OWT scheme is. The statement that 'Rrs is not compared directly..' and '3 optical variables' is critical, but the 3 variables should immediately be listed as it wasn't totally clear what those were in the following sentences. Recommend listing them directly as 1), 2) and 3) here. It appears that its OWT_area, AVW, and the NDI? Its unclear how many OWTs there are, and what their characteristics are in terms of the 3 parameters. A figure or table would be helpful.

**Response**: Thanks for pointing this out.

**Changes**: We have written a separate chapter on the OWT system and adapted previous information accordingly. We have also added an illustration of the mean spectra, the co-variances of the optical parameters and a brief description (new Fig. 4).

L293: Need references for 'fuzzy logic approach' or more description of its use.

**Changes**: This is integrated into the new separate chapter.

L326: Its unclear what is meant by 'the variability is broad' - environmental, optical, both? This whole sentence is a bit ambigous and phrasing is unclear: the sentence continues with 'possible problems of the water algorithms used', which now switches topic from optical variability (leading to?) higher uncertainties in optical products from applied algorithms (that is how I interpret this, which could be wrong).

**Changes**: We rephrased this: "Coastal and inland waters vary more in colour and are often dominated by optical effects from high concentrations of sediments, CDOM, or phytoplankton. This wide range of concentrations may also exceed the validity range of applied satellite algorithms."

L327: 'ambiguous variability' is ambiguous in itself. Do the authors mean, unpredictable Rrs shapes resulting from errors in atmospheric correction?

L328: '...with A4O having a possibly erroneous tendency towards more blue water (i.e. OWT 1)': this statement is unclear - I'm not sure what this means. Can you clarify and rephrase?

L329: ' The other AC methods would probably result...' - this sounds like conjecture - was this observed?

**Changes**: In response to the three points, we rephrased this section: "Hieronymi et al. (2023a) compared five atmospheric correction methods for OLCI and showed that, especially in the transition from coastal to clearer North Sea waters, the resulting $R_{rs}$ can be fundamentally different. Here, the comparative uncertainties in the shape and magnitude of $R_{rs}$ are particularly high, but also difficult to quantify. Comparison with in-situ data show a tendency for A4O to assess the water as clearer and bluer, i.e. to switch earlier than the other methods to maximum membership in OWT 1 during the transition to the open sea. Consequently, this means that absorption and scattering of water constituents are underestimated, and therefore the predicted concentrations are underestimated too."

L337: What is the reason for ascribing OWTs labeled with 'a' and 'b'?

**Response**: The OWT framework is very flexible and applicable to many band configurations. It focuses on three RGB bands to realize this flexibility and uses $R_{rs}$ shape and magnitude information. Of the ten defined classes, three of them have an additional label "a" or "b". They have the same spectral shape but different magnitudes (brightness). In this way, it is possible to define a separate class for coccolithophores for example (OWT 3b), calcifying phytoplankton with crucial function in global biogeochemical cycling. The IOPs of that single-celled algae are significantly different than of other algae groups. The approach allows therefore application of dedicated water algorithms, e.g. to estimate the related particulate inorganic carbon (PIC) concentration.

**Changes**: The OWT framework is better introduced in a separate section.

L344: Use of the word 'visible' twice in the same sentence for different reasons is confusing - suggest rephrasing.

**Changes**: One replaced with "recognisable".

L345: The underlying assumptions meaning that algorithms assume optically deep conditions?

**Response**: Yes, this is generally true for ocean/aquatic colour algorithms. Moreover, they expect moderate wind speed, certain sun height (if not at zenith), and a well-mixed upper layer without floating material.

**Changes**: Insert of clarifying words.

L347-348: By 'too blue', is the inference that pixels are erroneously classified? Its unclear what the OWTs are representing without a table, figure or other explanation.

**Response**: Within the new mentioned OWT section, we have included a figure that contains a table with OWT description.

L360: A frequency map of OWTs would illustrate the permanence or variability of areas in relation to OWTs (this could be a sub-figure to Fig. 10 or 11).

**Response**: This is an interesting point for future analysis, also the distribution of covarying class memberships. However, a frequency map of OWTs would be affected by inhomogeneous distributions of data gaps, which is the case because of diverse cloud cover (Fig. 3).

L383: 'not shown' implies this was tested, but the statement says '..it would be expected..' - this needs to be clarified as to whether in fact the authors did try this, or this is speculation or based on some other result which would need references in that case.

**Response**: The reviewer is right; the sentence is too speculative and is removed. The general behaviour was discussed in Hieronymi et al. (2023) and tested for the supporting information of Bi and Hieronymi (2024), but not specifically on the same period.

L390: Its not evident to the reader at this point what any OWTs look like or represent. This needs to be presented as a section devoted solely to the characteristics of the OWTs earlier in the manuscript.

**Response**: Within the new OWT section, we have included a figure that includes mean $R_{rs}$.

L396: This statement needs some references.

**Changes**: We added the following references: "(e.g., Nelson and Siegel 2013; Kutser et al., 2016; Spyrakos et al., 2018)".

L414-415: Are these global numbers or over the study area?

**Response**: This is unclear, as it says: "From global MODIS cloud observations, ...".

**Changes**: We include "worldwide land surfaces" to emphasize that this is not only our region of interest. Moreover, we include a reference for the following sentence: (e.g. Schrum et al., 2003).

L435: What does 'guaranteed' mean? This is a strong statement, and would recommend to modify.

**Changes**: We change it to "is satisfied on average".

L430-437: not sure if anything new or relevant from this section

**Response**: This is maybe not visible on the first view but contains interesting questions on validation. Wind speed is one factor that influences measured $R_{rs}$. Available matchups are significantly reduced if one applies a wind speed threshold of e.g. 6 m/s. At higher wind speeds, we may have optical effects from suspended particles or air bubbles in water, even causing a change of OWT distribution. Thus, we keep the section for future reference.

**References:**

Bi, S., Hieronymi, M., and Röttgers, R.: Bio-geo-optical modelling of natural waters, Front. Mar. Sci., 10, 1196352, https://doi.org/10.3389/fmars.2023.1196352, 2023.

Hieronymi, M., Bi, S., Müller, D., Schütt, E. M., Behr, D., Brockmann, C., Lebreton, C., Steinmetz, F., Stelzer, K., and Vanhellemont, Q.: Ocean color atmospheric correction methods in view of usability for different optical water types, Front. Mar. Sci., 10, 1129876, https://doi.org/10.3389/fmars.2023.1129876, 2023a.

Hieronymi, M., Röttgers, R., Alikas, K., Kratzer, S., Ansko, I., Behr, D., Bi, S., Burmester, H., Heymann, K., Novak, M., Pramlall, S., Rahn, I.-A., Roux, P., Thölen, C., and Voynova, Y. G.: Biogeo-optical and-chemical characterization of cyanobacterial blooms in the Baltic Sea for ocean colour satellite remote sensing, Alkor Cruise Report No. AL597, 24 pp., https://doi.org/10.3289/CR_AL597, 2023b.

Kutser, T., Paavel, B., Verpoorter, C., Ligi, M., Soomets, T., Toming, K., and Casal, G.: Remote sensing of black lakes and using 810 nm reflectance peak for retrieving water quality

parameters of optically complex waters. *Remote Sens.*, *8*(6), 497, https://doi.org/10.3390/rs8060497, 2016.

Nelson, N. B., and Siegel, D. A.: The global distribution and dynamics of chromophoric dissolved organic matter. *Annual review of marine science*, *5*(1), 447-476, https://doi.org/10.1146/annurev-marine-120710-100751, 2013.

Schrum, C., Hübner, U., Jacob, D., and Podzun, R.: A coupled atmosphere/ice/ocean model for the North Sea and the Baltic Sea. *Climate Dynamics*, *21*(2), 131-151, https://doi.org/10.1007/s00382-003-0322-8, 2003.

Spyrakos, E., O'Donnell, R., Hunter, P.D., Miller, C., Scott, M., Simis, S.G.H., Neil, C., Barbosa, C.C.F., Binding, C.E., Bradt, S., Bresciani, M., Dall'Olmo, G., Giardino, C., Gitelson, A.A., Kutser, T., Li, L., Matsushita, B., Martinez-Vicente, V., Matthews, M.W., Ogashawara, I., Ruiz-Verdú, A., Schalles, J.F., Tebbs, E., Zhang, Y., and Tyler, A.N.: Optical types of inland and coastal waters, Limnol. Oceanogr., 63, 846-870. https://doi.org/10.1002/lno.10674, 2018.

---

## Author Comment (AC4)

**CC1: 'Comment on essd-2025-443', Elizabeth C Atwood, 22 Dec 2025**

We would also like to thank you so much for your feedback and valuable comments. Our response and actions are listed below in blue.

**General comments:**

Excellent overview in the Introduction, albeit with some specific comments below. The method shows good promise but I have some reservations, noted in Specific Comments below. The speculative other features of the data (§5) are not especially unique to this dataset. The provided dataset, while impressive, is not validated, which is mentioned in the manuscript and should not forgotten by potential users (or in future references to this paper).

**Response**: Thank you. Especially regarding the question of validation, we would like to refer to our detailed responses to Reviewer #1 and our corresponding changes that include rough POC validation (for one water type) and explain difficulties of validation. Moreover, even though this end-to-end processing scheme is not yet published in detail or available code-wise, there are already some mentioned references that include validation, e.g. of $R_{rs}$ (Hieronymi et al., 2023).

**Specific comments**

L50-51: I don't seen a result in either reference saying that precipitation can be a source alone for CDOM?

**Response**: The reference mentioned Kieber et al. (2006) states in the abstract: All rainwater samples contained chromophoric dissolved organic matter (CDOM) as well as fluorescent compounds. This is what we also observed in our rainwater samples at the open Atlantic Ocean.

**Changes:** We added another relevant literature review to the diverse CDOM sources, namely Nelson and Siegel (2013).

L52: Water movement in rivers and lakes would be in a different category to currents, should be explicitly included since your main point is to cover the limnological to the oceanographic.

**Changes:** We rephrased the sentence to: "Sediments are kept in suspension by water movement in rivers and lakes, or in shallow waters by currents, tides, and waves."

L71-72: It is the aim to cover all water areas uniformly and with sufficient data quality, not only of this study but of these other services, no matter that there do still exist areas for improvement. I suggest removing "Thus".

**Changes:** OK, thanks – done.

L77: Compromised when a single AC method optimized for a specific water type is used.

**Changes:** We have revised the sentence as suggested to clarify this limitation. The text now reads: "... a requirement often compromised when a single AC method optimized for a specific water type is used."

L78-79: What about Atwood et al (2024, doi: 10.3390/rs16173267), which focused precisely on transitional water systems? Spyrakos et al (2018, doi: 10.1002/lno.10674) also focused on coastal as well as inland waters.

**Changes:** We acknowledge the relevance of these studies regarding transitional and coastal/inland water classification. We have added citations to Atwood et al. (2024) and Spyrakos et al. (2018) in the specified sentence to explicitly credit their contributions to this

domain: "This approach is particularly valuable for capturing transitional waters (e.g., river plume or algal bloom fronts), as also highlighted in studies of coastal and inland systems (Spyrakos et al., 2018; Atwood et al., 2024), where optical properties vary non-linearly with constituent concentrations."

L90-92: This is not as novel as it is being presented, such a system to continuously monitor from rivers, through coastal areas to the ocean has been demonstrated through the CERTO project.

**Changes:** We included a reference to the similar approach of the CERTO project and Atwood et al. (2024).

L113-116: This has good potential for large overrepresentation of more oceanic waters in the training dataset, thus skewing the cluster optimization results to be oriented on oceanic water conditions than the smaller by area transitional water systems.

**Response**: Here, we describe the sub-domain "coastal waters". The distance to coast is comparable with the high-resolution coastal product in the Copernicus Marine Service. We only define a sub-region that corresponds to the product and the general term coastal waters. We perform OWT analysis everywhere, but specifically look into this masked region too. The data points within this mask are not used for OWT training or clustering. Bi and Hieronymi (2024) is based on simulated data (Bi et al., 2023) and includes all optical extremes. The dataset can be used to focus on transitional water systems and OWT weight distributions, which is currently not discussed.

L144-146: To cite a method for the current study in a manuscript that is still being prepared is questionable.

**Response**: This is true, but the lengthy and complex development of the end-to-end processing scheme is also not easily publishable. Different aspects are published, like ONNS and the new OWT scheme. The atmospheric correction is similar as the one in C2RCC, which misses a proper reference too, but is widely used by the community. We rephrased the section.

L155-156: What was the range of time offsets between S-3A and 3B? For coastal systems which are strongly tidal, a difference of over 45 min during the right tidal cycle can greatly change the water optical properties in a single location. This is less of a worry for the Baltic Sea, but certainly for the North Sea, in particular the English Channel. I question then what a mean pixel-reflectance value represents when taken from images at different times/tidal conditions for those coastal regions.

**Response**: This is exactly the kind of feedback we hope to receive from users of the dataset – thank you very much. The masking needs to be carefully refined, among other things to highlight potential influences of sun glint (which is currently not done) or to identify areas affected by tidal dynamics. The exact time offset depends on the region. For the entire region the time window is approximately three hours of data acquisition.

**Changes:** We included a reference to Sent et al. (2025) on the importance of tides for ocean colour applications to estuaries. "So far, potential effects of tidal dynamics, particularly relevant in regions such as the Bristol Channel or the Elbe River estuary, have not been accounted for in the merging, which may influence the products (Sent et al., 2025)."

L173: With regard to the SST assumption for same latitude, not including elevation probably makes less of a difference over the study area, but in the next sentence you mention Dead Sea and Great Salt Lake. Surely on a global scale it would further be important to include additional

aspects (like elevation, distance from coast, etc) to estimate freshwater system SST, other than just latitude.

**Response**: The climatology used is under revision too, also because of your legitimate concerns. The climatological data assist the processing and in future more the flagging too, e.g. on tides. Nevertheless, improved bridging between the different Copernicus Services would also be desirable in this regard.

L283-284: The c-mean algorithm provides membership values to all classes, not a subset as suggested with "three to six classes usually contributing". If you aren't providing memberships to all classes, how do you determine what subset of classes to calculate memberships for?

**Response**: You are correct that the algorithm calculates membership values for all defined classes. Our statement referred to the practical observation that, for any given spectrum, typically only three to six classes exhibit significant membership probabilities (e.g., > 10e-4), while the remaining contributions are mathematically non-zero but negligible.

**Changes:** We have revised the text to be precise: "... During the OWT analysis of the reflectance, weights are assigned to all defined classes, although typically only three to six classes contribute significantly." Please also note that a new section and figure have been included according the comments of reviewer #2.

L285-286: A minimum requirement for total membership of 0.0001? I think this written differently than it was meant, that would be impossibly low for a summed membership even if unbounded. I also don't find "0.0001" in Hieronymi et al (2023a). I am going to assume this is meant a minimum threshold a valid dominant OWT membership, and I would argue this threshold is far too low. For bounded memberships, i.e. summing to 1, a good rule of thumb for minimum membership definition is 1/C where C is the total number of OWT classes. This would occur if a point is assigned equal membership to all clusters, thus suggesting it does not belong well to any cluster. This rule of course deviates in the case of unbounded memberships that can sum to more or less than 1, as in Moore et al. (2001), Jackson et al. (2017) and Bi & Hieronymi (2024). But it is argued in Bi & Hieronymi (2024) that memberships should still sum to close to 1 if there is no redundancy or underrepresentation in the classification results. Thus being somewhere close to 1/C threshold should still be advisable, which for the Bi & Hieronymi (2024) OWT class set with 10 classes, would be 0.1. Thus the threshold of 0.0001 seems unreasonably low to me. This point also relates to Fig. 12 and conclusions made therefrom.

**Response**: The reviewer is correct that for standard Fuzzy C-Means where memberships sum to 1, a threshold of 0.0001 would be meaningless. However, our method (Bi and Hieronymi, 2024) follows the approach of Moore et al. (2001), where individual class memberships represent probabilities based on the Mahalanobis distance. These are unbounded and their sum (Utot) indicates the overall similarity of the spectrum to the training dataset.
Here, Utot acts as a "novelty detector" or outlier index. A threshold of 1/C (0.1) would be appropriate for identifying ambiguous classifications in a closed set, but for outlier detection in our probabilistic framework, it would be too restrictive and exclude valid waters that deviate slightly from the class centroids. The threshold of 10e-4 is empirically chosen to flag only extreme outliers (e.g., severe glint, clouds, or non-water targets). This specific threshold value is documented in Table 3 of Hieronymi et al. (2023a) and on page 6 of Hieronymi et al. (2017).

**Changes:** We have added these references to the text for clarity: "The total membership serves as an indicator of the quality of the classifiability; a minimum requirement of 0.0001 is often used (e.g. Moore et al., 2001; Hieronymi et al., 2017, 2023a)."

L343-345: High variability of OWT class could come from other aspects besides shallow water and visible seafloor – unless that is also changing greatly? This variability could also be due to a location in a narrow water connection between to large water bodies. This also relates to the conclusion on L389 regarding using OWT classes as a mask for optically shallow water areas – is this supposed to follow over the high variability in OWT class signal?

**Response**: The main point here is that up to now, we have no real handle to mask out optically shallow water. However, in case of the island mentioned (or coral island in the ocean), surrounding water should be comparable. Comparisons with high resolution Sentinel-2 images show sandy bottom effects, partly with dark sea grass areas that cause different OWT allocation. This is also visible on some lake boundaries. During the year, adjacency effects may also occur or change with land cover (up to now assumed to be small for A4O) or sub-pixel contamination of growing plants may interfere, which could cause OWT variability. These points are already mentioned.

L360-361: Point well taken that low optical variability over time is important for ground-truthing and SVC, but the point in the parentheses "but rather for clear waters" is confusing. Do you mean that only clearer waters are important for ground truth and SVC? I would argue that optically complex water, but with low variability, would also be important for these efforts so as to cover a larger portion of the full spectrum.

**Response**: We agree with the reviewer that low temporal variability is the primary requirement for reliable reference data. While operational SVC typically relies on oligotrophic waters (to minimize Lw contributions and isolate atmospheric path radiance), we agree that stable, optically complex waters are scientifically valuable. They are particularly critical for ground-truthing (validation) to assess algorithm performance across the full dynamic range of the sensor.

OWT technology offers the opportunity to apply SVC per water type and per product. But some classes cover extremes that occur in small areas (black lakes, OWT 7) or small-time scales like intense phytoplankton bloom (OWT 5b). Moreover, your mentioned point with tidal effects might be relevant, as mentioned in the Bristol Channel, with always dominating OWT 6. Adjacency effects, shallow water, high cloud cover (rainy areas, sensor pollution), etc. should be avoided.

**Changes:** We have revised the text to differentiate these needs while highlighting the importance of stability for both. Revised text: "… For ground-truthing and definition of system vicarious calibration (SVC) gains, low optical variability over time is more important than water clarity alone. While SVC is traditionally performed in clear waters, stable optically complex waters are essential for validating algorithms and ground-truthing across the full spectral range. OWT technology provides the opportunity to apply SVC on a per–water-type and per-product basis."

L447-449: It is at the moment speculative if OWT serve to better characterize aquatic carbon, including the example OWT 3b bloom, and this should be phrased accordingly.

**Response**: In fact, OWT narrows down ranges of water constituents and covariances of them. In the example with POC, there are linear contributions by plankton and detritus with different weight. Separation of both is essential for the magnitude of POC. Moreover, we can apply OWT-specific biogeo-optical relationships like for POC and DOC. The sources of DOC can be different in the diverse aquatic environments.

**Changes:** We reformulated "Optical water types have the potential to support improved characterization of aquatic carbon."

L504-506: Mélin & Vantrepotte (2015) and Spyrakos et al. (2018) both focus specifically on coastal/transitional water systems, thus your conclusion is not fully supported. Also a critique that 17 classes may be too many simply on the basis of lacking in-situ data is not a very strong argument.

**Response:** We accept the reviewer's correction. We acknowledge that the cited frameworks (e.g., Mélin & Vantrepotte, 2015; Spyrakos et al., 2018) indeed cover coastal and transitional waters. Regarding the number of classes, our argument was intended to address the statistical robustness of the validation process rather than the availability of data per se. Dividing limited in-situ matchups into a high number of fine-scale classes (e.g., 17) often results in insufficient sample sizes for reliable performance assessment per class. We have revised the text to correct the description of previous studies and clarify this motivation.

**Changes:** Revised text: "... Alternatively, other OWT frameworks can serve as a basis (e.g. Moore et al., 2001; Vantrepotte et al., 2012; Moore et al., 2014; Mélin and Vantrepotte, 2015; Jackson et al., 2017; Spyrakos et al., 2018; Bi et al., 2021; Atwood et al., 2024). However, some of these frameworks differentiate a large number of optical classes (e.g., up to 17). While scientifically rigorous, such high granularity can be challenging for operational validation, where a more consolidated set of classes is advantageous to ensure sufficient matchup density for robust statistical assessment of each water type."

**Technical corrections**

L55: The sentence starting here rather belongs thematically to the next paragraph.

**Changes:** A paragraph has been added here.

L100-101: You should indicate the location of Glasgow in Fig 1 if you reference it in the text. Same goes for the Elbe River catchment.

**Changes:** In response to another suggestion of reviewer #2, Fig. 1 now contains labels of all mentioned areas. Mentioning Glasgow as location has been removed.

L130: What do the various boxes (white or red) represent? This is not explained in the figure caption.

**Changes:** The boxes of individual scenes (white) and of the region of interest (red) have been explained in the figure caption. Moreover, Fig. 2 contains now a workflow.

L184: Suggested rewording for clarity: "Products from climatologies and their derivatives, such as white cap fraction, refer..."

**Changes:** Thank you, included.

L200: The first three parameters (OWT_AVW, OWT_Area, OWT_NDI) are rather spectral curve characteristics directly from the Rrs, that are not dependent on output from an OWT classification? And OWT_index, is this the dominant OWT class?

**Response:** The optical variables are part of the OWT classification. Index is indeed for the class with maximum membership.

**Changes:** "Parameters associated with the optical water type classification".

L278: Is the AVW meant from Vandermeulen et al 2020? Then this shouldn't be mixed up with a normal weighted mean, better to change to "weighted harmonic mean" so this remains clear.

**Response**: The hyperspectral AVW in Vandermeulen et al. 2020 is calculated for the range 400–700 nm, our AVW refers to 400 to 800 nm to include hyper-eutrophic cases.

**Changes:** In response to reviewer #2, we introduced the OWT framework with a separate section and overview figure and rephrased this content.

L560: Remove the either "to" or "must", incorrect with both.

**Changes:** Thanks, we removed "to".

**References:**

Atwood, E. C., Jackson, T., Laurenson, A., Jönsson, B. F., Spyrakos, E., Jiang, D., Sent, G., Selmes, N., Simis, S., Danne, O., Tyler, A., and Groom, S.: Framework for Regional to Global Extension of Optical Water Types for Remote Sensing of Optically Complex Transitional Water Bodies. Remote Sens., 16(17), 3267. https://doi.org/10.3390/rs16173267, 2024.

Hieronymi, M., Bi, S., Müller, D., Schütt, E. M., Behr, D., Brockmann, C., Lebreton, C., Steinmetz, F., Stelzer, K., and Vanhellemont, Q.: Ocean color atmospheric correction methods in view of usability for different optical water types, Front. Mar. Sci., 10, 1129876, https://doi.org/10.3389/fmars.2023.1129876, 2023a.

Kieber, R. J., Whitehead, R. F., Reid, S. N., Willey, J. D., and Seaton, P. J.: Chromophoric dissolved organic matter (CDOM) in rainwater, southeastern North Carolina, USA, J. Atmos. Chem., 54, 21-41, https://doi.org/10.1007/s10874-005-9008-4, 2006.

Nelson, N. B., and Siegel, D. A.: The global distribution and dynamics of chromophoric dissolved organic matter. *Annual review of marine science*, *5*(1), 447-476, https://doi.org/10.1146/annurev-marine-120710-100751, 2013.

Sent, G., Antunes, C., Spyrakos, E., Jackson, T., Atwood, E. C., and Brito, A. C.: What time is the tide? The importance of tides for ocean colour applications to estuaries. Remote Sensing Applications: Society and Environment, 37, 101425., https://doi.org/10.1016/j.rsase.2024.101425, 2025.

Spyrakos, E., O'Donnell, R., Hunter, P.D., Miller, C., Scott, M., Simis, S.G.H., Neil, C., Barbosa, C.C.F., Binding, C.E., Bradt, S., Bresciani, M., Dall'Olmo, G., Giardino, C., Gitelson, A.A., Kutser, T., Li, L., Matsushita, B., Martinez-Vicente, V., Matthews, M.W., Ogashawara, I., Ruiz-Verdú, A., Schalles, J.F., Tebbs, E., Zhang, Y., and Tyler, A.N.: Optical types of inland and coastal waters, Limnol. Oceanogr., 63, 846-870. https://doi.org/10.1002/lno.10674, 2018.